# Surveillance of cohesin-supported chromosome structure controls meiotic progression

Maikel Castellano-Pozo[1], Sarai Pacheco [1], Georgios Sioutas[1], Angel Luis Jaso-Tamame[1], Marian H. Dore[1], Mohammad M. Karimi [1] & Enrique Martinez-Perez [1,2 ✉]

Chromosome movements and programmed DNA double-strand breaks (DSBs) promote homologue pairing and initiate recombination at meiosis onset. Meiotic progression involves checkpoint-controlled termination of these events when all homologue pairs achieve synapsis and form crossover precursors. Exploiting the temporo-spatial organisation of the *C. elegans* germline and time-resolved methods of protein removal, we show that surveillance of the synaptonemal complex (SC) controls meiotic progression. In nuclei with fully synapsed homologues and crossover precursors, removing different meiosis-specific cohesin complexes, which are individually required for SC stability, or a SC central region component causes functional redeployment of the chromosome movement and DSB machinery, triggering whole-nucleus reorganisation. This apparent reversal of the meiotic programme requires CHK-2 kinase reactivation via signalling from chromosome axes containing HORMA proteins, but occurs in the absence of transcriptional changes. Our results uncover an unexpected plasticity of the meiotic programme and show how chromosome signalling orchestrates nuclear organisation and meiotic progression.

[1] MRC London Institute of Medical Sciences, London W12 0NN, UK. [2] Imperial College Faculty of Medicine, London W12 0NN, UK. ✉email: fadri@imperial.ac.uk

The formation of haploid gametes from diploid germ cells during meiosis constitutes a cornerstone of sexual reproduction. Defects in this process cause sterility and aneuploid gametes that impair fitness of the resulting progeny. Accurate transmission of chromosomes into the gametes depends critically on the establishment of crossover (CO) events between paternal and maternal homologous chromosomes (homologues) during the long prophase preceding the first meiotic division, when COs, together with sister chromatid cohesion (SCC), ensure correct chromosome orientation on the spindle[1]. Thus, meiotic cells have evolved meiosis-specific chromosome structures and surveillance mechanisms to ensure that every pair of homologues is connected by COs before nuclei proceed to the first meiotic division.

Meiotic chromosome morphogenesis requires the assembly of proteinaceous axial elements containing meiosis-specific versions of cohesin, the complex that provides SCC. In yeast loading of cohesin-containing Rec8, a meiosis-specific version of cohesin's mitotic kleisin Scc1[2], ensures proper meiotic chromosome function. Higher eukaryotes use additional meiosis-specific kleisins, including Rad21L in mouse and the highly identical COH-3 and COH-4 in *Caenorhabditis elegans*[3–5]. In addition to establishing SCC, loading of meiotic cohesin promotes recruitment of HORMA-domain proteins (HORMADs) to axial elements, rendering chromosomes competent to initiate meiotic recombination, via the formation of DNA double-strand breaks (DSBs), and to undergo pairing of homologous chromosomes. Homologue pairing is also facilitated by cytoskeleton-driven chromosome movements during early meiotic prophase and culminates with the assembly of the synaptonemal complex (SC), a ladder-like structure that bridges together the axial elements of aligned homologues[6]. This process, known as synapsis, stabilises homologue interactions and is essential to ensure that a subset of DSBs become CO-designated sites during the pachytene stage of meiotic prophase, which is defined by full synapsis. Thus, the processes leading to CO formation are closely integrated with the establishment of meiosis-specific chromosome structures built over a cohesin scaffold. However, how different cohesin complexes and the SC contribute to meiotic chromosome function once full synapsis is achieved and CO precursors are formed remains poorly understood.

The formation and repair of DSBs into COs is mechanistically coupled to early meiotic progression by a network of surveillance mechanisms that monitor specific, but incompletely understood, pairing and recombination intermediates[7]. The presence (or absence) of these meiotic chromosome metabolism intermediates results in signals that feedback to regulate checkpoint kinases, which then target components of the pairing, recombination, and cell cycle machineries[8]. These quality control mechanisms fulfil two important roles. Firstly, they act to limit the temporal window during which nuclei remain competent for DSB formation, ensuring timely cessation of this activity once CO precursors are formed on all chromosomes, thus preventing the genotoxic effects of excess DSBs. Secondly, they regulate meiotic progression in a checkpoint manner, inducing arrest of nuclei at the stage when defects in specific CO-promoting events are first detected. For example, mutations that impair DSB processing into CO precursors cause extension of DSB-permissive stages in yeast and *C. elegans*[9–11]. The regulated exit from DSB-permissive stages represents a fundamental transition of the meiotic programme, but how nucleus-wide loss in competence for DSB formation is sustained remains unclear.

The *C. elegans* germline provides a powerful system to investigate how feedback mechanisms integrate pairing and recombination with early meiotic progression[12]. CHK-2 promotes DSB formation, chromosome movement, and SC assembly during early prophase and the temporal window of CHK-2 activity is controlled by feedback from the progression of these events

mediated by HORMADs[13–15]. Despite sharing some components, these feedback mechanisms are mechanistically different from checkpoints that induce apoptosis of nuclei with persistent DNA damage or asynapsed chromosomes at late pachytene[16,17]. Importantly, SC assembly is independent of recombination in worms[18] and both processes are under surveillance to feedback on CHK-2[19]. In wild-type germlines synapsis induces termination of CHK-2-dependent chromosome movements by pachytene entrance, while the formation of CO precursors triggers loss of CHK-2-dependent markers of DSB formation by mid pachytene. Once achieved, nucleus-wide loss of CHK-2 activity is thought of as a unidirectional transition of the meiotic programme that leads to the completion of recombination and progression towards chromosome segregation.

In the current work, we combine the experimental advantages of the *C. elegans* germline with temporally resolved protein removal methods to investigate how cohesin and the SC contribute to meiotic chromosome function once full synapsis and early recombination steps are completed. We uncover a role for REC-8 and COH-3/4 cohesin in promoting SC stability and demonstrate that direct or indirect, via cohesin removal, SC disassembly triggers rapid and nucleus-wide redeployment of the pairing and recombination machinery, inducing drastic changes in nuclear organisation. This apparent reversal in meiotic progression requires CHK-2 reactivation mediated by HORMA protein HTP-1, but occurs in the absence of transcriptional changes. Our findings have important implications for understanding the quality control mechanisms that ensure fertility in higher eukaryotes.

## Results

**Time-resolved removal of specific cohesin complexes from pachytene nuclei.** *C. elegans* expresses two types of meiosis-specific cohesin complexes defined by their kleisin subunit: REC-8[4] and the highly identical (84%) and functionally redundant COH-3 and COH-4[3] (referred to as COH-3/4 from now on). Both REC-8 and COH-3/4 are prominent components of pachytene axial elements and mutant analysis has uncovered overlapping roles for REC-8 and COH-3/4 cohesin in axis assembly, as well as non-overlapping roles in SCC, pairing, synapsis, and recombination[3,4,20,21]. Crucially, however, *rec-8* and *coh-3/4* mutants display severe chromosome organisation defects from the onset of meiosis, making mutant analysis impractical for clarifying the roles of REC-8 and COH-3/4 complexes during pachytene. To bypass this intrinsic limitation of kleisin mutant analysis, we created kleisin versions that can be removed from meiotic chromosomes in a temporally controlled manner by introducing three repeats of the TEV protease recognition motif in REC-8 (after Q289) and COH-3 (after I315). TEV-mediated cleavage of kleisin subunits mimics cleavage by separase at anaphase onset, and this approach has been successfully used to rapidly remove cohesin from chromosomes in vivo in other organisms[22,23]. Both REC-8[3XTEV]::GFP and COH-3[3XTEV]::mCherry were fully functional, as single-copy transgenes expressing REC-8 and COH-3 from their respective promoter and 3′ UTRs complemented the meiotic defects of *rec-8* and *coh-3 coh-4* mutants, respectively (Supplementary Fig. 1a, b). Unless otherwise indicated, TEV experiments described below were performed in germlines from worms expressing REC-8[3XTEV]::GFP or COH-3[3XTEV]::mCherry in backgrounds carrying null mutations in *rec-8* or *coh-3 coh-4*, respectively.

The sequential stages of meiotic prophase are easily identified in the *C. elegans* germline based on the position of nuclei within the germline, nuclear appearance, and by cytological markers of specific meiotic events (Fig. 1a). Nuclei in leptotene–zygotene

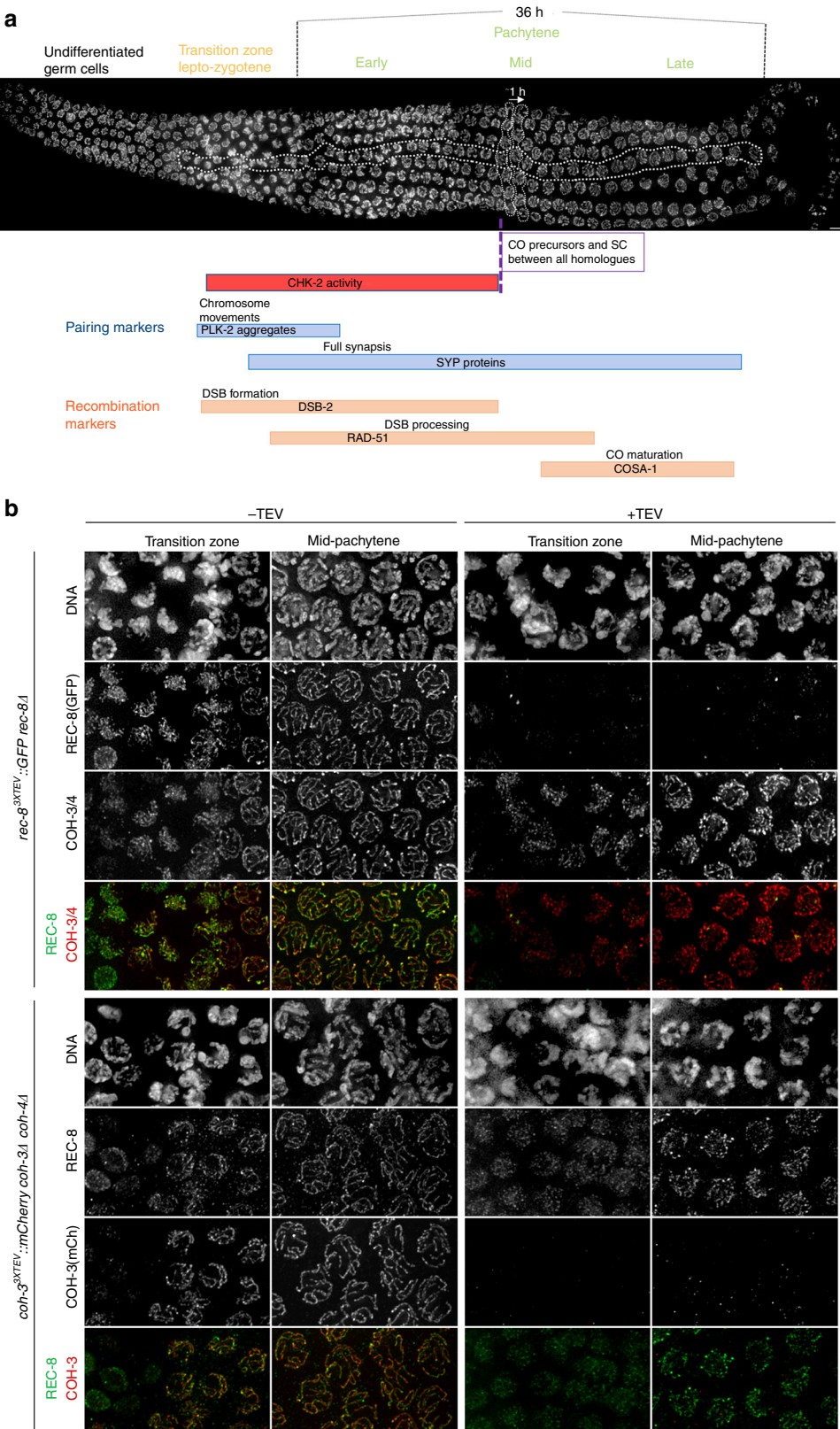

**Fig. 1 TEV-mediated cohesin removal in the _C. elegans_ germline. a** Example of a dissected wild-type _C. elegans_ germline stained with DAPI indicating key meiotic events and the timing of appearance and disappearance of different cytological markers of pairing and recombination. The horizontal dotted line encloses 34 nuclei progressing from leptotene to late pachytene. Nuclei move horizontally along the germline at a rate of ~1 row per hour, as indicated by two vertical rows of nuclei enclosed by dotted lines. REC-8 and COH-3 localisation in the same germline is shown in Supplementary Fig. 1e. **b** Projections of nuclei at the indicated stages before and after (3.5 h) TEV injection demonstrating efficient and specific removal of REC-8[3XTEV]::GFP and COH-3[3XTEV]:: mCherry. Data are representative of two independent experiments. Scale bar = 5 µm. See also Supplementary Fig. 1.

stages (transition zone) are characterised by chromosome clustering and by markers of chromosome movement, SC assembly, and DSB formation. Entry into pachytene is marked by chromosome dispersal, high numbers of early recombination intermediates, and full SC assembly. By late pachytene, markers of early recombination disappear and CO-designated sites emerge. Since germ cells in *C. elegans* are syncytial, we reasoned that microinjection of the TEV protease into the germline of live worms would allow us to remove REC-8[3XTEV]::GFP or COH-3[3XTEV]::mCherry from the large population of nuclei at the pachytene stage. Indeed, germlines dissected and fixed 3.5 h after TEV injection demonstrated efficient removal of REC-8[3XTEV]:: GFP or COH-3[3XTEV]::mCherry from axial elements at all stages of meiotic prophase while non-targeted complexes remained bound to chromosomes (Fig. 1b and Supplementary Fig. 1d, e). Importantly, nuclei take about 36 h to progress through pachytene[24], migrating at a rate of approximately one cell row per hour[25] (Fig. 1a). Therefore, germline dissection 3.5 h post TEV injection allowed us to directly address the functional requirement of REC-8 and COH-3/4 cohesin at all substages of pachytene. TEV injection in control germlines expressing REC-8:: GFP or COH-3::mCherry without the TEV motif confirmed that

only TEV-tagged versions are removed following injection (Supplementary Fig. 1c). Thus, this TEV-based approach provides a powerful tool to rapidly and specifically remove different cohesin complexes from pachytene chromosomes.

**Removal of REC-8 or COH-3/4 cohesin induces SC disassembly.** Having set up conditions to rapidly remove cohesin from chromosomes, we investigated the contribution of REC-8 and COH-3/4 complexes to axial elements and the SC in pachytene nuclei, as these structures are assembled over a cohesin scaffold during leptotene–zygotene. The central region of the SC in *C. elegans* is composed of four proteins (SYP-1–4) that are interdependent for their loading during early prophase[26–29]. Using anti-SYP-1 antibodies we observed that removal of REC-8[3XTEV]::GFP or COH-3[3XTEV]::mCherry caused SC disassembly at all stages of meiotic prophase (Fig. 2a and Supplementary Fig. 2a). In contrast, the SC remained intact when COH-3[3XTEV]::mCherry was removed from pachytene nuclei of worms expressing the wild-type version of COH-4 from the endogenous locus (Supplementary Fig. 2b), consistent with the functional redundancy of COH-3/4[3]. These results show that REC-8 and COH-3/4 are individually required for SC stability in pachytene nuclei.

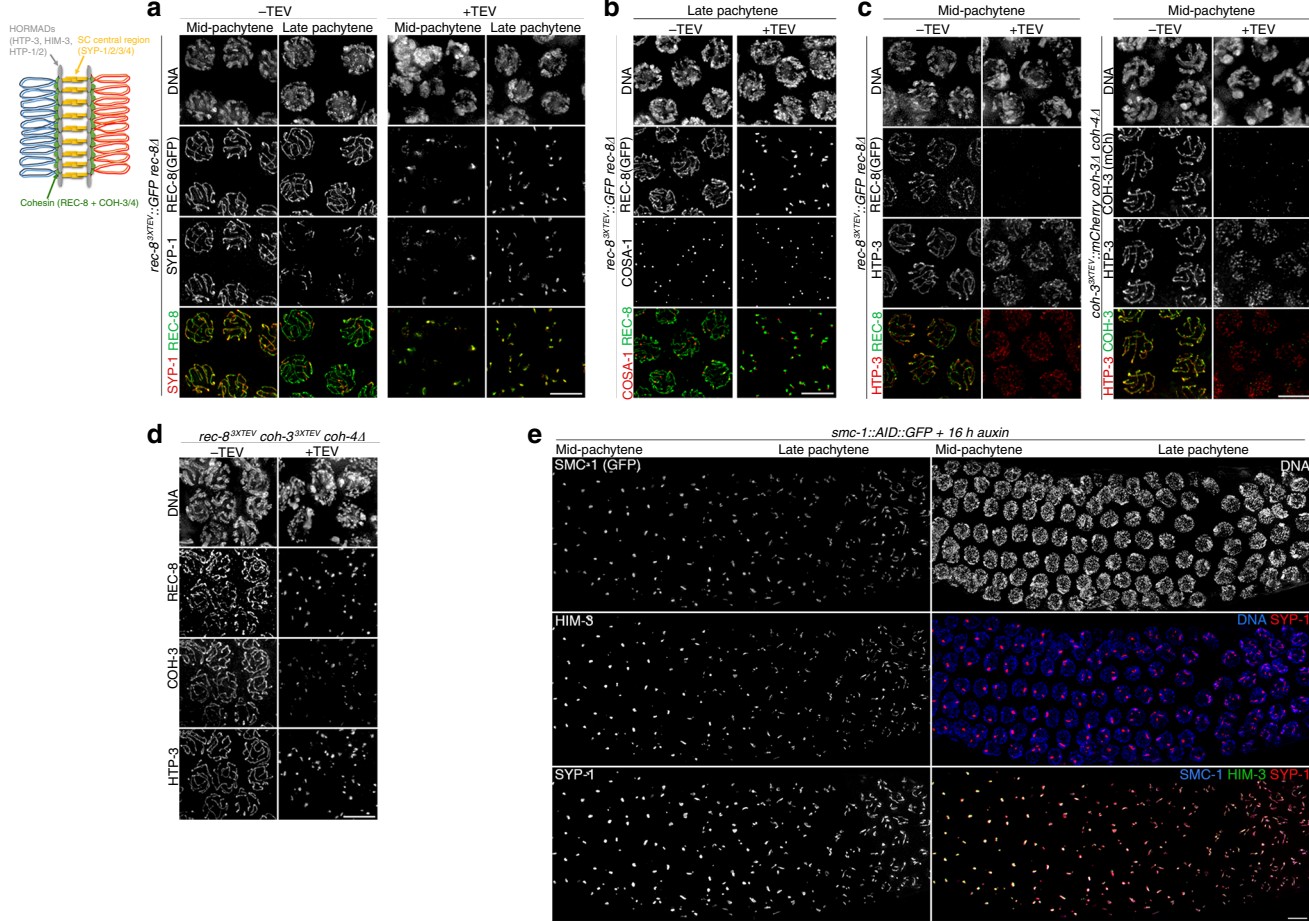

**Fig. 2 Effect of cohesin removal in the chromosomal structures of pachytene nuclei.** Diagram of a pair of synapsed homologues at the pachytene stage indicating the localisation of axis and SC components. **a–d** Contain projections of nuclei at the indicated stages before and after (3.5 h) TEV injection acquired on a Delta Vision microscope, while **e** shows nuclei after auxin degron-mediated depletion of SMC-1::AID:GFP acquired with a SIM microscope. **a** TEV-mediated removal of REC-8[3XTEV]::GFP induces SC disassembly. **b** COSA-1 foci persist following TEV-mediated removal of REC-8[3XTEV]::GFP. **c** Axial elements containing HTP-3 persist following TEV-mediated removal of REC-8[3XTEV]::GFP or COH-3[3XTEV]::mCherry. Note that HTP-3 signals appear more discontinuous when COH-3[3XTEV]::mCherry is removed. **d** Simultaneous depletion of REC-8[3XTEV] and COH-3[3XTEV] results in axial element disassembly. **e** Auxin degron-mediated depletion of SMC-1::AID::GFP for 16 h induces axis and SC disassembly. Data are representative of two independent experiments. Scale bar = 5 μm in all panels. See also Supplementary Fig. 2.

In most germlines, REC-8[3XTEV]::GFP cleavage caused concentration of SYP-1 in a few aggregates per nucleus, where it colocalized with remaining GFP (REC-8) signals (Fig. 2a). These aggregates were more prominent in late pachytene nuclei, where they typically appeared as 6 chromosome-associated short stretches (Fig. 2a). Since late pachytene nuclei in worms contain six CO-designated sites, one per homologue pair, that are visible as 6 COSA-1 foci[30], we tagged the cosa-1 gene with HA using CRISPR and crossed this allele into the REC-8[3XTEV]::GFP strain. Following TEV injection, we observed that 92% of persistent REC-8[3XTEV]::GFP stretches in late pachytene were associated with a COSA-1 focus and that most nuclei contained 6 COSA-1 foci (Fig. 2b and Supplementary Figs. 2c, d). These observations suggest that REC-8 cohesin is locally regulated around CO-designated sites.

**Disassembly of axial elements requires simultaneous removal of REC-8 and COH-3/4.** Axial elements in *C. elegans* require HORMAD HTP-3, which is recruited to chromosomes at meiotic onset in a cohesin-dependent manner[3,31,32]. HTP-3 then acts as a scaffold for the recruitment of additional HORMADs: HIM-3 and HTP-1/2, which play multiple roles in pairing, synapsis, recombination, and SCC release during the meiotic divisions[33–36]. All four HORMADs localise along the whole length of axial elements until late pachytene, when COs trigger restricted removal of HTP-1/2[37]. In contrast to SC central region components, all HORMADs remained associated with axial elements following cleavage of REC-8[3XTEV]::GFP or COH-3[3XTEV]::mCherry (Fig. 2c and Supplementary Figs. 2e–g). We noted that staining of HORMADs became weaker and more discontinuous following removal of COH-3[3XTEV]::mCherry than REC-8[3XTEV]::GFP, suggesting that COH-3/4 cohesin makes a larger contribution to axis integrity in pachytene nuclei, in agreement with the weaker axis appearance observed in *coh-3 coh-4* double mutants when compared to *rec-8* mutants[3]. To confirm that both types of cohesin promote axis integrity in pachytene nuclei, we used CRISPR to introduce the 3XTEV motifs in *rec-8* and *coh-3* in a strain carrying a *coh-4* null mutation. TEV injection in these germlines induced efficient removal of both REC-8 and COH-3 from axial elements (Fig. 2d) and caused loss of HTP-3 tracks, confirming that both REC-8 and COH-3 cohesin promote axis stability in pachytene. We further tested this by depleting SMC-1, which should affect all types of cohesin, in pachytene nuclei using the auxin degron system[38]. This method induced slower and less penetrant cohesin depletion from axial elements compared with the TEV approach. After 8 h of auxin treatment we observed partial SMC-1 removal from axial elements, but importantly regions lacking SMC-1 also lacked HIM-3 and SYP-1, confirming a local requirement for cohesin in axis and SC stability (Supplementary Fig. 2h, i). After 16 h of auxin treatment we observed full disassembly of axial elements and the SC throughout pachytene, including nuclei in mid–late pachytene that had entered the pachytene stage before worms were exposed to auxin (Fig. 2e). Interestingly, we noticed that short tracks of SMC-1 that persisted in late pachytene nuclei after 16 h of auxin treatment overlapped with COSA-1 foci (Supplementary Fig. 2j). These results are consistent with REC-8 and COH-3/4 cohesin contributing independently to axis stability in pachytene nuclei and provide further evidence that cohesin is locally regulated around CO sites.

**Cohesin removal reactivates chromosome movement and DSB formation in pachytene nuclei.** CHK-2 acts as a master regulator of early prophase events in *C. elegans*[13]. Its activation at meiotic onset triggers DSB formation and chromosome movements that play a central role in homologue pairing and that induce chromosome clustering, giving chromatin a characteristic crescent shape appearance (Fig. 1a, b). In wild-type germlines, chromosome clustering is restricted to nuclei in the transition zone, corresponding to leptotene/zygotene, as pachytene entry is marked by chromosome dispersal (Fig. 1a). However, DAPI staining of germlines dissected 3.5 h post REC-8 or COH-3 removal revealed nuclei with clustered chromosomes throughout most of the pachytene region (Fig. 3a–d and Supplementary Figs. 3a–f), suggesting de novo establishment of chromosome movement. To confirm this, we used molecular markers of chromosome-end attachment to the nuclear envelope, a widely conserved feature of early prophase that in worms requires CHK-2 and pairing centre-binding (PCB) proteins that localise to one end of each chromosome[12]. CHK-2 triggers recruitment of Polo-like kinase 2 (PLK-2) to PCB proteins by phosphorylating their Polo-binding domain, PLK-2 then promotes aggregation of SUN-1/ZYG-12 on the nuclear envelope to initiate cytoskeleton-driven chromosome movements[39–42]. In wild-type germlines, nuclei undergoing chromosome movements in transition zone display multiple PLK-2 aggregates, which disappear as nuclei progress into pachytene. Since a single PLK-2 aggregate associated with the paired X chromosomes often persists into pachytene until all chromosomes form CO precursors[19], we used the presence of more than one PLK-2 aggregate as an indicator of active chromosome movement. As indicated in Fig. 1a, the germline can be divided into vertical rows of nuclei progressing from leptotene to late pachytene, with each germline typically containing around 40 such vertical rows. In control germlines, nuclei with multiple PLK-2 aggregates correspond to ~30% of the total number of vertical rows of nuclei progressing from leptotene to late pachytene (Fig. 3a, b). In contrast, 3.5 h post TEV-induced removal of REC-8[3XTEV]::GFP or COH-3[3XTEV]::mCherry, nuclei with PLK-2 aggregates occupied over 70% of meiotic prophase, including most of the pachytene region (Fig. 3a, b and Supplementary Fig. 3a, b). We observed the same situation using antibodies against HIM-8 pT64, which label CHK-2-dependent phosphorylation of all PCB proteins at their Polo-binding domain[15], confirming reacquisition of this CHK-2 marker in pachytene nuclei (Supplementary Figs. 3c, d). To determine if de novo formation of PLK-2 aggregates upon cohesin removal was functional, we monitored the presence of PLK-2-dependent SUN-1 S12 phosphorylation, which is normally restricted to transition zone nuclei[39,40]. Similar to our observations above, removal of REC-8[3XTEV]::GFP or COH-3[3XTEV]::mCherry induced de novo appearance of SUN-1 pS12 in pachytene nuclei (Supplementary Figs. 3e, f). Therefore, removal of REC-8 or COH-3/4 from pachytene chromosomes induces rapid reacquisition of CHK-2-dependent markers of chromosome movement.

We next asked whether cohesin removal would also induce redeployment of the DSB machinery in pachytene nuclei. During early prophase, CHK-2 promotes association of DSB-1/2 with chromosomes, two factors required for DSB formation and whose association with chromosomes indicates a DSB-permissive state[9,10]. In control germlines, DSB-2 staining was observed in transition zone and early–mid-pachytene nuclei (Fig. 3c, d). Removal of REC-8[3XTEV]::GFP or COH-3[3XTEV]::mCherry caused a clear increase in the percentage of rows of prophase nuclei positive for DSB-2 (Fig. 3c, d), confirming de novo association of DSB-2 with pachytene chromosomes and suggesting reacquisition of DSB competence. Finally, we investigated the effect of REC-8 removal on RAD-51 foci, which label early recombination intermediates produced upon resection of SPO-11 DSBs[27]. REC-8 removal induced increased RAD-51 foci at all prophase stages (up to late pachytene), and this increase was dependent on SPO-11, as RAD-51 foci were largely absent when REC-8 was removed in a *spo-11* mutant background (Fig. 3e and Supplementary Fig. 3g).

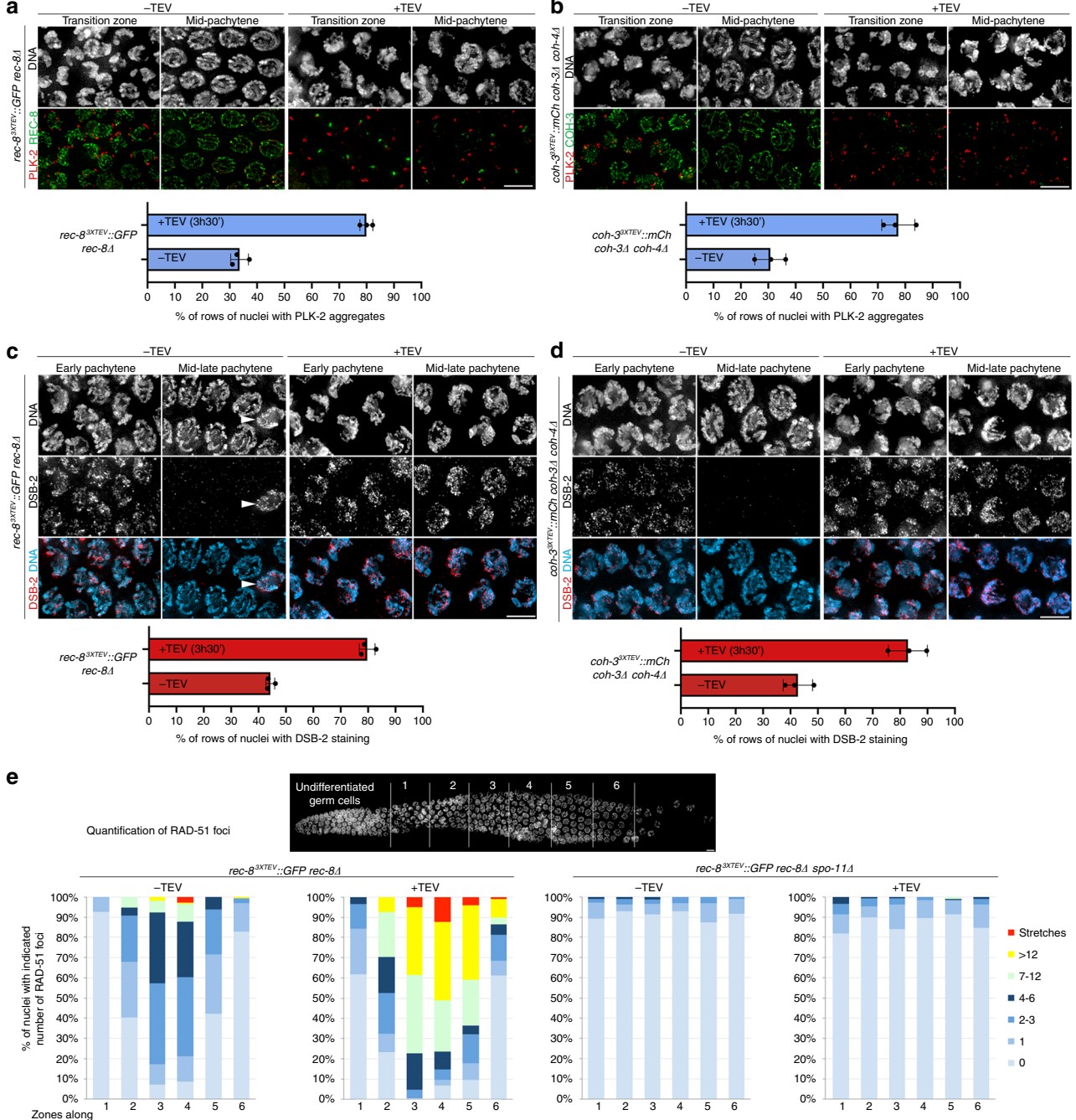

**Fig. 3 Cohesin removal induces reappearance of chromosome-movement and DSB-formation markers in pachytene nuclei. a, b** TEV-mediated removal of REC-8[3XTEV]::GFP or COH-3[3XTEV]::mCherry induces reappearance of PLK-2 aggregates on the nuclear envelope. Quantification indicates percentage of vertical rows of nuclei (see Fig. 1a) between meiotic onset and end of pachytene in which over 50% of nuclei contained more than 1 PLK-2 aggregate on the nuclear envelope. **c, d** TEV-mediated removal of REC-8[3XTEV]::GFP or COH-3[3XTEV]::mCherry induces reappearance of DSB-2 in pachytene nuclei. Arrow head on **c** indicates an arrested nucleus that retains DSB-2 in the pachytene region. Quantification indicates percentage of vertical rows of nuclei between meiotic onset and end of pachytene in which over 50% of nuclei were positive for DSB-2. **e** Quantification of RAD-51 foci per nucleus before and after TEV-mediated removal of REC-8[3XTEV]::GFP in wild-type and *spo-11* mutant germlines (three germlines scored per genotype, see Supplementary Table 4 for number of nuclei quantified per zone). The region between meiosis onset and the end of pachytene was divided into six zones as indicated in the DAPI-stained germline. Note that TEV injection increases RAD-51 foci in pachytene nuclei of wild-type, but not *spo-11* mutant, germlines. Scale bar = 5 μm in all panels. Three germlines were quantified for graphs in **a–d**, data are presented as mean values ± standard deviations. See also Supplementary Fig. 3. Source data for graphs in all panels are provided as Source Data file.

The increase in RAD-51 foci and the redeployment of DSB-2 to pachytene chromosomes suggests that REC-8 removal induces de novo DSB formation in pachytene nuclei. We cannot exclude that the increase in RAD-51 foci can be partially explained by a

requirement of REC-8 cohesin in ongoing repair of DSBs generated before TEV injection.

These results demonstrate that pachytene nuclei with full SC and a normal complement of CO precursors retain the potential

to reactivate early, CHK-2-dependent, prophase events and suggest that chromosome-bound cohesin plays an important role in regulating CHK-2 activity during pachytene.

**Direct SC removal from pachytene nuclei induces rapid reactivation of chromosome movement and DSB formation.** The observation that cohesin removal from pachytene chromosomes triggers SC disassembly and reactivation of CHK-2-dependent events, led us to ask whether direct SC removal from pachytene chromosomes would on its own cause CHK-2 reactivation. We used the auxin-inducible system[38] to attempt time-controlled degradation of the SC central region component SYP-2 in pachytene nuclei as we expected that removal of one component would be sufficient to destabilise the SC. Four hours after young adult worms were placed on auxin plates, SC tracks, visualised with anti-SYP-1 antibodies, had disappeared from all pachytene nuclei (Fig. 4a). Moreover, DAPI staining revealed reacquisition of chromosome clustering throughout most of the pachytene region, suggesting CHK-2 reactivation. Staining with anti-PLK-2 antibodies confirmed this, as over 70% of nuclei between leptotene and late pachytene displayed multiple PLK-2 aggregates in germlines from auxin treated worms, compared to 39% in untreated controls (Fig. 4b and Supplementary Fig. 4a). In vivo imaging of worms expressing mScarlet::SYP-3 and PLK-2::GFP showed that SYP-2 depletion for just 2 h triggered SC disassembly and reappearance of PLK-2 aggregates in pachytene nuclei (Fig. 4c and Supplementary Movies 1–3). De novo formed PLK-2::GFP aggregates displayed extensive movement on the nuclear envelope, including fusion and splitting events characteristic of transition zone nuclei (Supplementary Movie 4), and their average speed was not different from that observed in pachytene nuclei of *syp-2* mutants (Fig. 4d and Supplementary Movies 5 and 6). Moreover, X-chromosome pairing centre regions were associated following direct (*syp-2::AID*) or indirect (COH-3[3XTEV] cleavage) SC disassembly (Supplementary Fig. 4b, c), consistent with chromosome movements promoting SC-independent pairing of these regions, as observed in mutants lacking SC[26].

SC depletion also triggered de novo association of DSB-2 with pachytene chromosomes (Supplementary Fig. 4d), and increased RAD-51 foci (Fig. 4e), consistent with de novo DSB formation. Interestingly, COSA-1 foci in late pachytene nuclei remained intact following complete SC disassembly (Supplementary Fig. 4e), consistent with previous reports that used 1,6-hexanediol to dissolve the SC in dissected germlines[43]. Therefore, SC removal is sufficient to induce CHK-2 reactivation in pachytene nuclei, suggesting that SC surveillance remains active once the structure is fully assembled.

**Limited SC disassembly triggered by cohesin removal is sufficient for reactivation of CHK-2-dependent events.** To further clarify the interplay between cohesin, SC stability, and CHK-2 reactivation we used different approaches to induce partial cohesin depletion from pachytene nuclei. First, we imaged *rec-8[3XTEV]::GFP* germlines that were dissected and fixed just 1.5 h after TEV injection, instead of 3.5 h as in previous experiments. Most mid-pachytene nuclei from germlines dissected 1.5 h post TEV injection displayed partial REC-8 removal and limited SC disassembly, but already displayed clear reappearance of PLK-2 aggregates associated with chromosomal ends (Supplementary Fig. 5a). Second, we monitored PLK-2 aggregates and the SC in germlines from *smc-1::AID::GFP* worms following 8 h of auxin treatment, which induced partial cohesin removal (Supplementary Fig. 2i). These germlines also displayed partial SC disassembly in mid-pachytene nuclei, which showed robust reappearance of PLK-2 aggregates (Supplementary Fig. 5b–d).

Third, we also used the auxin degron system to deplete REC-8::AID::GFP from pachytene nuclei. Auxin treatment of these worms resulted in slower and less complete REC-8 removal compared with the TEV approach. Four hours of auxin treatment induced partial loss of REC-8 and the SC from early pachytene nuclei, which displayed reappearance of PLK-2 aggregates (Supplementary Fig. 5f, g). By 8 h of auxin treatment, PLK-2 aggregates also accumulated in mid-pachytene nuclei (Supplementary Fig. 5g). These results suggest that the SC is locally destabilised when cohesin is removed from chromosomes and that this limited SC disassembly is sufficient to reactivate CHK-2 in a nucleus-wide fashion. Interestingly, in all three experimental approaches described above SC disassembly was much more pronounced in early pachytene nuclei than in mid and late pachytene nuclei, consistent with recent findings demonstrating the SC becomes more stable as nuclei progress through pachytene[44–46].

**Reinstallation of the DSB-formation and chromosome-movement machinery in pachytene nuclei requires CHK-2 and HORMAD HTP-1.** Our observations thus far show that cohesin and SC depletion from pachytene nuclei trigger reactivation of early steps of pairing and recombination, which under unchallenged conditions are restricted to early prophase stages and are dependent on CHK-2 activity. Thus, we sought to confirm whether the reinstatement of early prophase events seen in pachytene nuclei following cohesin removal occurs via CHK-2 activation. As *chk-2* mutants accumulate severe meiotic defects from the onset of meiosis, we tested whether the auxin system could be used to induce rapid CHK-2 depletion. Homozygous *chk-2::AID* worms (generated by CRISPR) displayed normal chiasma formation despite slower SC assembly (Supplementary Figs. 6a, b). Importantly, following 6 h of auxin treatment we observed complete loss of PLK-2 aggregates (Fig. 5a) and DSB-2 staining (Fig. 5b) from germlines of *chk-2::AID* worms, consistent with loss of CHK-2 activity. Therefore, we treated *chk-2::AID* worms with auxin for 6 h before inducing TEV-mediated REC-8[3XTEV]::GFP removal and evaluating the presence of PLK-2 aggregates on the nuclear envelope (Fig. 5c). Injection of the TEV protease following CHK-2 depletion resulted in REC-8 removal (Fig. 5c) and SC disassembly (Fig. 5e) from pachytene chromosomes, but failed to induce reappearance of PLK-2 aggregates on the nuclear envelope (Fig. 5c), as observed in control REC-8[3XTEV]::GFP germlines (Fig. 5d). Similarly, TEV-mediated removal of REC-8[3XTEV]::GFP following CHK-2 depletion failed to induce reappearance of DSB-2 in pachytene nuclei (Supplementary Fig. 6c). These results confirm that CHK-2 is required for reimplementing early prophase events in pachytene nuclei following REC-8 removal and SC disassembly.

HORMAD HTP-1 is a key component of the feedback mechanisms that couple SC assembly with meiotic progression during early prophase[14,36]. HTP-1 promotes persistence of CHK-2-dependent chromosome clustering in mutant backgrounds in which SC assembly fails in one or more chromosomes[14,47], presumably by participating in the creation and transmission of a signal that sustains CHK-2 activity in the presence of unsynapsed chromosomes. To investigate if HTP-1 is required for the reactivation of CHK-2-dependent events observed when cohesin is removed from pachytene chromosomes, we induced TEV-mediated removal of COH-3 in germlines of *htp-1* mutants. TEV injection induced efficient COH-3 removal in *htp-1* mutant germlines, but failed to induce reappearance of PLK-2 aggregates or chromosome clustering in pachytene nuclei (Fig. 5f), as observed when COH-3 was removed in control germlines (Fig. 5g). This suggests that the structural changes that result when cohesin is removed from pachytene chromosomes are

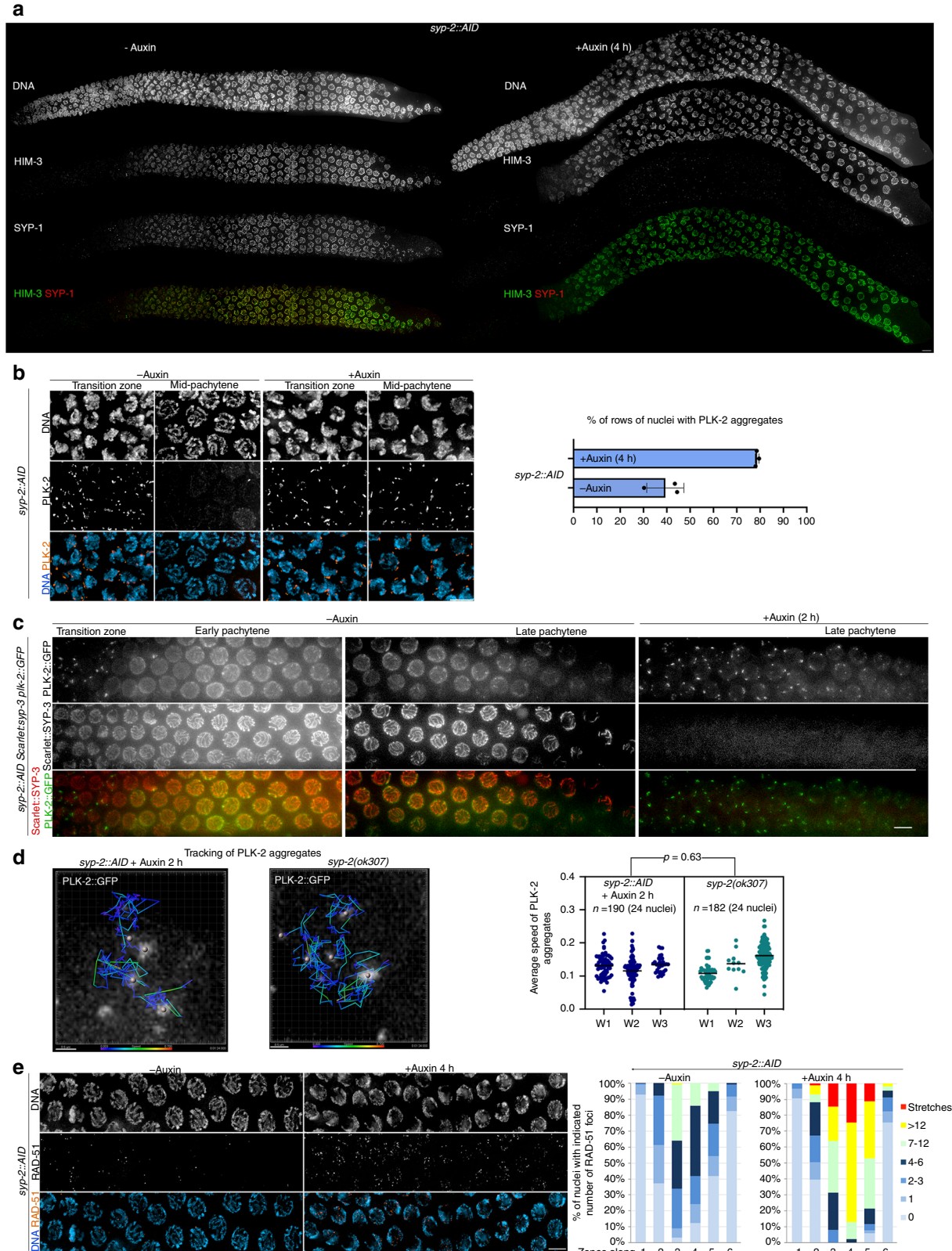

sensed and transmitted by the same, HTP-1-dependent, surveillance mechanisms that monitor SC assembly to regulate CHK-2 activity during early prophase.

**CHK-2 depletion causes rapid disassembly of the DSB-formation and chromosome-movement machinery.** During

the control experiments to validate the depletion of CHK-2 using the auxin degron, we observed that auxin treatment for 6 h induced complete loss of PLK-2 aggregates and DSB-2 staining from transition zone and early pachytene nuclei (Fig. 5a, b). This suggested that CHK-2 is not simply required at the onset of prophase to kick-start pairing and recombination, but rather that

**Fig. 4 SC removal induces reappearance of chromosome-movement and DSB-formation markers in pachytene nuclei. a** Auxin treatment of *syp-2::AID* worms induces rapid SC disassembly (visualised with anti-SYP-1 antibodies) without affecting axial elements (visualised with anti-HIM-3 antibodies). **b** Auxin treatment (4 h) of *syp-2::AID* worms induces reappearance of PLK-2 aggregates on the nuclear envelope of pachytene nuclei. Quantification indicates percentage of vertical rows of nuclei between meiotic onset and the end of pachytene in which over 50% of nuclei were positive for PLK-2 aggregates (three germlines per genotype, data are presented as mean values ± standard deviations). **c** In vivo imaging of germlines from *syp-2::AID* worms expressing mScarlet::SYP-3 (SC) and PLK-2::GFP. Note SC disassembly and reappearance of PLK-2::GFP aggregates following 2 h of auxin treatment. Data are representative of three independent experiments. See also Supplementary Movies 1–4 and Supplementary Fig. 4. **d** Tracking of PLK-2::GFP aggregates in pachytene nuclei of *syp-2:AID* (+Auxin 2 h) and *syp-2(ok307)* mutants over 7 min at 5 s intervals. Quantification of the average speed of PLK-2::GFP aggregates in 24 nuclei filmed from three different worms (W1-3) shows no significant differences (*P* values calculated by a two-tailed Nested *t* test) between *syp-2::AID* (+Auxin 2 h) and *syp-2(ok307)*. Supplementary Movies 5–6 show tracking of PLK-2 aggregates in the two nuclei shown in this panel. **e** Auxin treatment (4 h) of *syp-2::AID* worms induces accumulation of RAD-51 foci in pachytene nuclei. Quantification of RAD-51 foci per nucleus before and after auxin-induced SC removal (three germlines scored per genotype, see Supplementary Table 4 for number of nuclei quantified per zone). Scale bar = 5 μm in **b**, **c**, and **e**. Source data for graphs in **b**, **d**, and **e** are provided as Source Data file.

sustained CHK-2 activity during early prophase is required to prevent premature disassembly of the pairing and recombination machinery. We further tested this by depleting CHK-2 from *syp-1* RNAi germlines, which accumulate nuclei with PLK-2 aggregates and chromosome clustering throughout most of the pachytene region due to impaired synapsis from the onset of meiosis. Indeed, CHK-2 depletion for only 3 h triggered complete loss of PLK-2 aggregates and chromosome clustering from all prophase nuclei (Fig. 5h), confirming that persistence of PLK-2 aggregates and chromosome clustering require sustained CHK-2 activity. These results suggest that activation and inactivation of CHK-2 rapidly regulates the localisation of the pairing and recombination machinery, whose components remain responsive to CHK-2 status through most of pachytene.

**CHK-2 reactivation switches back early meiotic events in the absence of transcriptional changes.** The drastic changes in nuclear organisation observed when cohesin or the SC are removed from pachytene nuclei, which include reappearance of chromosome-associated markers of DSB formation and chromosome movement, led us to ask whether reestablishment of these events involved transcriptional changes of the corresponding genes. Thus, we performed whole-worm RNAseq to compare the transcriptome before and after 4 h of auxin-mediated depletion of SYP-2::AID, which induced rapid accumulation of nuclei with CHK-2-dependent markers in the pachytene region (Fig. 4b and Supplementary Fig. 4d), focusing our analysis on a list of 335 meiotic genes[12,48] (Supplementary Data 1). We detected no significant changes in the expression of meiotic genes (including *dsb-2* and *chk-2*) following SYP-2 depletion, but observed increased expression of the proapoptotic factor *egl-1* (Fig. 6a), suggesting that SYP-2 depletion triggers an apoptotic response, consistent with the increased levels of apoptosis observed in *syp-2* mutants[27]. Therefore, the striking changes in nuclear organisation caused by direct SC disassembly occur in the absence of obvious transcriptional changes of genes known to be involved in meiosis, suggesting that CHK-2 substrates remain present in pachytene nuclei and that their localisation is modulated according to the status of CHK-2 activity. In agreement with this, we found that despite the large increase in chromosome-associated DSB-2 signal triggered by SYP-2::AID depletion (Supplementary Fig. 4d), overall protein levels of DSB-2 remained unchanged before and after SYP-2 removal (Fig. 6b). Due to technical constrains we have not investigated the transcriptional status of meiotic genes following TEV-mediated removal of COH-3 or REC-8 cohesin. Based on our findings with the *syp-2* degron, we infer that nuclear reorganisation caused by REC-8 or COH-3 removal is not driven by transcriptional changes of meiotic genes.

**Cohesin-containing axial elements contribute to maintain CHK-2 activation.** While performing experiments involving the simultaneous removal of COH-3[3XTEV] and REC-8[3XTEV] shown in Fig. 2d, we noticed that reacquisition of chromosome clustering in pachytene nuclei 3.5 h post TEV injection was less pronounced than when REC-8 or COH-3 were individually removed. This suggested that the integrity of cohesin-containing axial elements is important to fully reactivate CHK-2, or to sustain this activation. In fact, while we consistently observed the formation of multiple PLK-2 aggregates on the nuclear periphery of pachytene nuclei when REC-8 or COH-3 were individually removed (Fig. 3a, b), or when SYP-2 was depleted (Fig. 4b), simultaneous removal of REC-8[3XTEV] and COH-3[3XTEV] resulted in many early pachytene nuclei displaying a single PLK-2 aggregate associated with the nuclear periphery (Fig. 7a). To clarify if cohesin-containing axial elements are required to sustain high levels of CHK-2 activity, we performed simultaneous removal of COH-3[3XTEV] and REC-8[3XTEV] in germlines of *syp-1* RNAi worms, in which impaired synapsis induces accumulation of nuclei with CHK-2 markers through most of the pachytene region. The efficiency of REC-8[3XTEV] and COH-3[3XTEV] removal was confirmed by the disappearance of HTP-3 tracks. As expected, control (*syp-1* RNAi) germlines displayed extensive accumulation of nuclei with multiple PLK-2 aggregates, while TEV injection caused a clear reduction in the number of these nuclei and the appearance of nuclei with 1 or no PLK-2 aggregates on the nuclear envelope (Fig. 7b and Supplementary Fig. 7). Similarly, the reappearance of PLK-2 aggregates on the nuclear envelope induced by partial SC disassembly in *smc-1::AID* worms treated with auxin for 8 h (Supplementary Fig. 5b–d) was lost when *smc-1::AID* worms were exposed to auxin for 16 h (Supplementary Fig. 5e), which caused complete axis and SC disassembly (Fig. 2e) and colocalization of PLK-2 with SYP-1 aggregates (Supplementary Fig. 5e). These results confirm that the integrity of axial elements is required to sustain full CHK-2 activation.

**Late pachytene nuclei are unable to reactivate CHK-2 upon SC or cohesin removal.** In all the experiments described above we noted that cohesin removal (REC-8[3XTEV] or COH-3[3XTEV] individually removed) or SC depletion consistently induced reappearance of CHK-2 markers through most of the pachytene region, except for very late pachytene nuclei. For example, we typically observed that 70–80% of nuclei between leptotene and late pachytene became positive for PLK-2 aggregates and DSB-2 staining following cohesin removal or SC depletion, with the remaining 20–30% of nuclei lacking these CHK-2-dependent markers invariably corresponding to the vertical rows of nuclei situated in late pachytene (Figs. 3a–d and 4b, and Supplementary Fig. 4d). In fact, simultaneous visualization of PLK-2 aggregates and COSA-1 foci following REC-8 removal or SC depletion

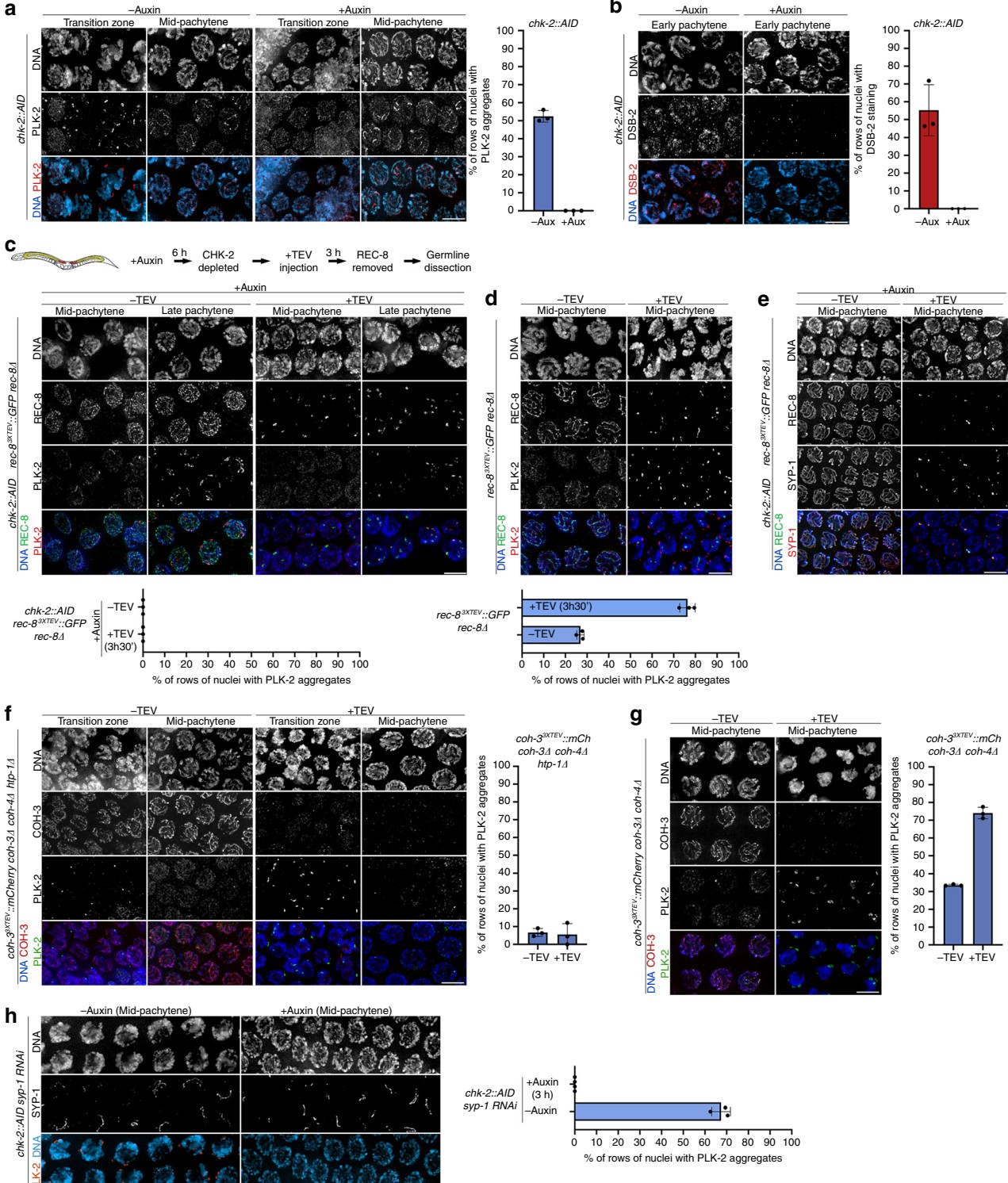

revealed few nuclei displaying both markers (Supplementary Fig. 8a, b). As SC depletion in late pachytene nuclei occurs as efficiently as in earlier stages (Fig. 4a), the inability to reactivate CHK-2-dependent events during late pachytene suggests that nuclei at this stage lose the ability to respond to the presence of unsynapsed chromosomes.

## Discussion
Our findings reveal that nuclei with fully synapsed chromosomes and a normal complement of CO precursors undergo rapid,

nucleus-wide, reestablishment of chromosome movement and DSB formation when SC stability is compromised. This process requires de novo CHK-2 activation and causes a dramatic reorganisation of the nucleus that appears to functionally revert pachytene nuclei back to earlier prophase stages, revealing a striking plasticity of the meiotic programme. We identify SC surveillance as the mechanism that regulates CHK-2 activity during pachytene. We also uncover a role for REC-8 and COH-3/4 cohesin in promoting SC stability in pachytene nuclei, as removal of either type of cohesin causes rapid SC disassembly and

**Fig. 5 CHK-2 is required for reactivating chromosome movement and DSB formation in pachytene nuclei.** Auxin treatment (6 h) of *chk-2::AID* worms induces disappearance of PLK-2 aggregates (**a**) and DSB-2 (**b**) from all stages of meiotic prophase. **c, d** Diagram of experimental design for auxin-induced CHK-2 depletion before TEV-mediated REC-8[3XTEV]::GFP removal. Note that in CHK-2-depleted germlines PLK-2 aggregates are not reformed following REC-8[3XTEV]::GFP removal (**c**), while REC-8[3XTEV]::GFP removal induces reappearance of PLK-2 aggregates in pachytene nuclei (**d**). **e** TEV-mediated removal of REC-8[3XTEV]::GFP in CHK-2-depleted germlines induces SC disassembly. Data are representative of two independent experiments. TEV-mediated removal of COH-3[3XTEV]::mCherry in *htp-1* mutants fails to induce reappearance of PLK-2 aggregates in pachytene nuclei (**f**), as observed in wild-type germlines (**g**). **h** Auxin-mediated depletion of CHK-2::AID (3 h) induces disappearance of PLK-2 aggregates from pachytene nuclei of *syp-1 RNAi* germlines. Graphs indicate percentage of vertical rows of nuclei between meiotic onset and end of pachytene in which over 50% of nuclei contained more than 1 PLK-2 aggregate on the nuclear envelope (**a**, **c**–**h**), or DSB-2 staining (**b**). Scale bar = 5 μm in all panels. Three germlines were quantified for graphs in all panels (except four germlines in **h** + auxin), data are presented as mean values ± standard deviations. Source data for graphs in **a**–**c**, **d**, **f**–**h** are provided as Source Data file.

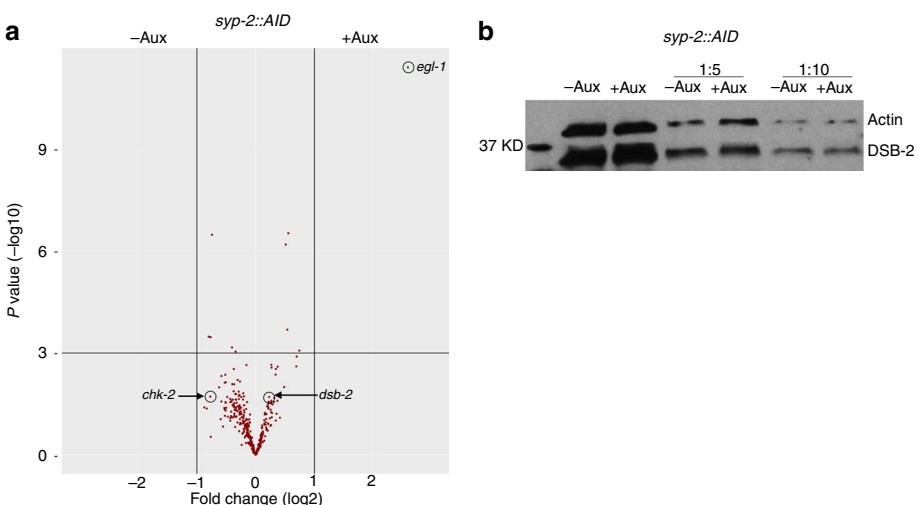

**Fig. 6 SC depletion does not trigger transcriptional changes of meiotic genes. a** Volcano plot showing transcriptional changes (whole-worm RNAseq) of 335 meiotic genes (Supplementary Data 1) in *syp-2::AID* worms before and after (4 h) auxin treatment. Note that *egl-1* is the only gene that appears significantly upregulated with a fold change higher than 2 (*P* values were calculated by a two-tailed Wald test and adjusted for multiple testing using the Benjamini and Hochberg method). **b** Western blot with extracts (1:1, 1:5, and 1:10 dilution) from *syp-2:AID* worms before and after treatment showing that the amount of DSB-2 protein is not increased following SYP-2 depletion. Actin is used as loading control. Data are representative of two independent experiments. Source data for blot in **b** is provided as Source Data file.

CHK-2 reactivation. This role of REC-8 and COH-3/4 (Rad21L) cohesin is different from their redundant role in promoting axis assembly at the onset of meiosis[3,49], evidencing that the contribution of cohesin to landmark meiotic chromosome structures (axial elements and the SC) is more complex than previously thought. Moreover, we also provide evidence that cohesin may be locally regulated around CO sites, reminiscent of the local regulation of SC structure at CO sites[50]. Below, we discuss our findings in the context of a model in which SC surveillance orchestrates meiotic progression by continuously regulating CHK-2 activity (Fig. 8).

In *C. elegans* CHK-2 promotes chromosome movement, DSB formation, and SC assembly at meiotic entrance[13] and CHK-2 activity is lost gradually; markers of chromosome movement disappear at pachytene entrance, coinciding with full synapsis, while markers of DSB formation disappear at mid pachytene, presumably once all chromosomes have formed CO precursors[9,10,19]. Analysis of mutants with defects in synapsis or CO formation suggest that the termination of chromosome movement at early pachytene and of DSB formation at mid pachytene, which are controlled by checkpoint mechanisms[9,10,14,19], represent two key functional transitions of meiotic progression that culminate with a nucleus-wide loss in competency for CHK-2-dependent events. However, we now show that nuclei that have progressed normally to mid pachytene undergo rapid reactivation of CHK-2-dependent events when SC stability is compromised. This finding provides important insights into the surveillance mechanisms that regulate meiotic progression.

First, the fact that mid-pachytene nuclei retain full potential for CHK-2 reactivation evidences that the kinase remains present in the nucleus in an inactive, but responsive, status. In principle, CHK-2 inactivation could be achieved by inhibitory signals triggered when all chromosome pairs form CO precursors, or by the extinction of a positive signal that promotes CHK-2 activity until all chromosomes achieve synapsis and form CO precursors. Our findings support the latter hypothesis and suggest that the status of synapsis plays a key role in extinguishing such signal (see below). We propose that meiotic prophase can be divided into two broad functional stages: one spanning between leptotene and late pachytene in which signalling from SC surveillance determines the level of CHK-2 activity, and a second starting in late pachytene in which CHK-2 is permanently shut down independently of SC status (Fig. 8). CHK-2 shut down at late pachytene coincides with a remodelling of axial elements that includes the partial removal of HTP-1 from axial elements[37], the onset of SC disassembly[51], and changes in the mode of DSB repair[52], suggesting that all these events are mechanistically coupled. The "CHK-2 responsive stage" can be further divided into three phases according to the level of CHK-2 activity present in the nucleus. During leptotene/zygotene, high CHK-2 activity promotes chromosome movement, DSB formation, and SC assembly. Then, by early pachytene, once SC assembly is completed, CHK-2 activity decreases to levels that sustain DSB formation but not chromosome movement. Finally, by mid pachytene, once CO precursors are present on all chromosomes CHK-2 activity

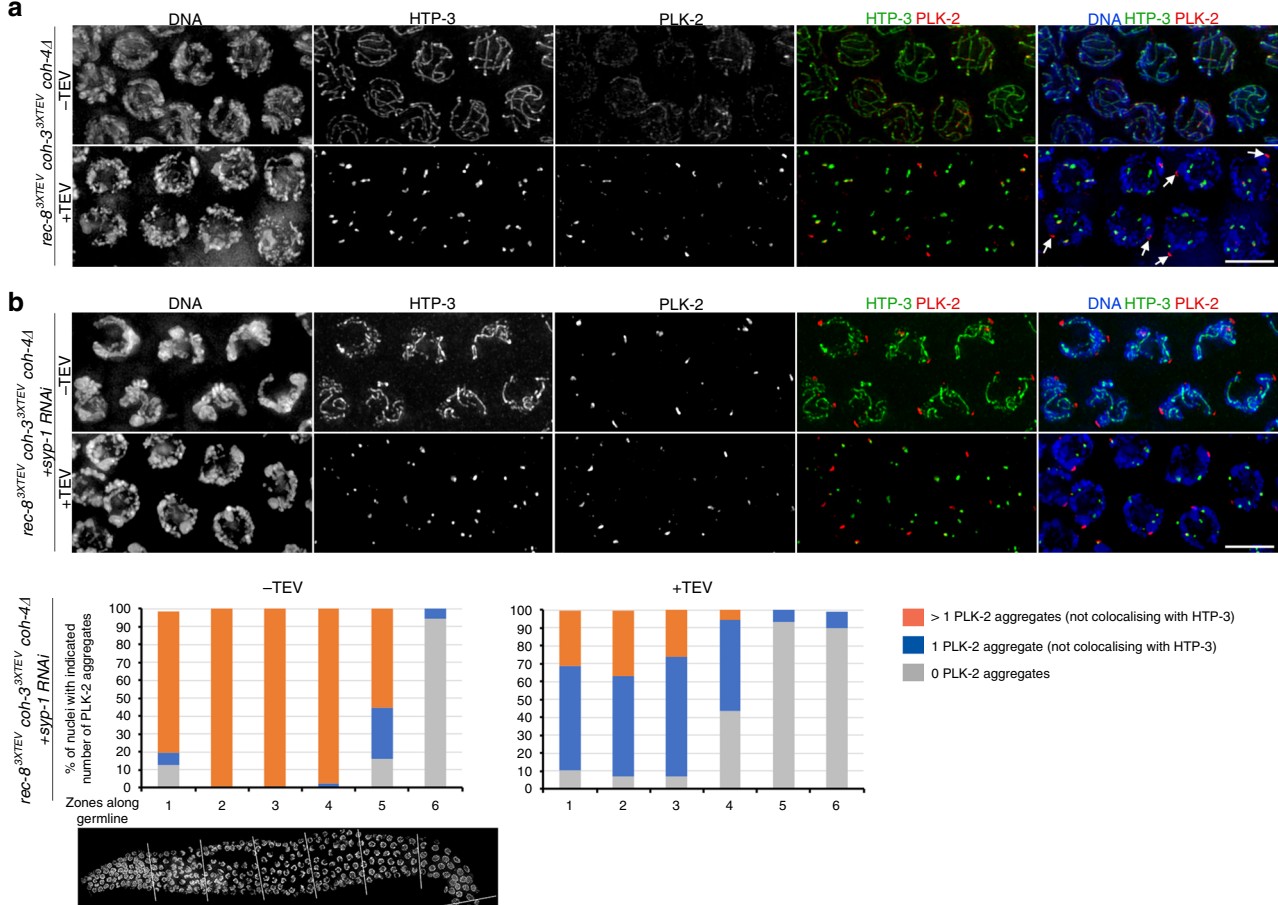

**Fig. 7 Axial elements contribute to sustain CHK-2 activity. a** Simultaneous removal of REC-8[3XTEV] and COH-3[3XTEV] induces limited formation of PLK-2 aggregates on the nuclear envelope (not colocalising with nuclear aggregates of HTP-3, indicated by arrows). Compare to images in Fig. 3a, b where either REC-8 or COH-3 are removed and multiple PLK-2 aggregates form on the nuclear envelope of pachytene nuclei. Data are representative of two independent experiments. **b** Simultaneous removal of REC-8[3XTEV] and COH-3[3XTEV] from *syp-1 RNAi* germlines reduces the number of PLK-2 aggregates on the nuclear envelope. Graphs show quantification of the percentage of nuclei with indicated number of PLK-2 aggregates, germlines were divided into six equal-size regions between meiotic onset and the end of pachytene and three germlines were scored for each condition. Data are representative of two independent experiments. Scale bar = 5 µm in all panels. Source data for graphs in **b** are provided as Source Data file. See also Supplementary Fig. 6.

abates, but nuclei remain competent for CHK-2 reactivation. We propose that similar to Plk1 during mitosis in mammalian cells[53], specific CHK-2-dependent functions may require different thresholds of kinase activity.

Second, our results uncover the presence of strong CHK-2-counteracting activities in prophase nuclei, which were evident by the rapid disappearance of markers of chromosome movement and DSB formation following CHK-2 depletion. This CHK-2 antagonistic activity is likely executed by phosphatases that dephosphorylate CHK-2 substrates to promote disassembly of the chromosome movement and DSB machineries. In addition, phosphatases could also antagonise CHK-2 directly by promoting dephosphorylation of activating sites, as seen in other organisms[54]. Identifying new CHK-2 substrates will be important to elucidate how the activity of CHK-2 and opposing phosphatases is integrated with the progression of pairing and recombination.

Third, de novo assembly of the chromosome-movement and DSB-formation machinery in the absence of transcriptional changes suggests that components of these complex processes remain present in the nucleus and are responsive to CHK-2 status until late pachytene. How specific substrates are affected by nucleus-wide levels of CHK-2 activity probably depends on complex interactions between CHK-2 and opposing phosphatases, which activity and subnuclear location are also likely to be highly regulated, as observed in the case of mitotic kinases and phosphatases[55].

Fourth, our findings suggest that throughout the "CHK-2 responsive stage" the level of nucleus-wide CHK-2 activity is largely determined by positive feedback signals emitted from chromosomes. HORMADs bound to axial elements, which can monitor in situ the status of pairing and recombination, are obvious candidates for signal generation[15], while a soluble pool of these proteins could also participate in the transmission of the signal to generate a nucleus-wide response[47]. Our finding that HTP-1 is required for reactivation of chromosome movement following cohesin removal provides further support for this model. How CHK-2 is activated in *C. elegans* remains unclear, as it lacks the N-terminal SQ/TQ cluster that in other organisms is phosphorylated by ATM and ATR to induce CHK-2 activation[54], but it may involve dimerisation and autophosphorylation in trans as observed in yeast Mek1[56]. CHK-2 deactivation could be mediated by phosphorylation removal at activating sites, or by phosphorylation of inactivating sites at its FHA domain, as observed in mammalian Chk2[57]. Elucidating how signals from chromosomes and the activity of phosphatases are integrated to implement nucleus-wide control of CHK-2 remains an important goal for future studies.

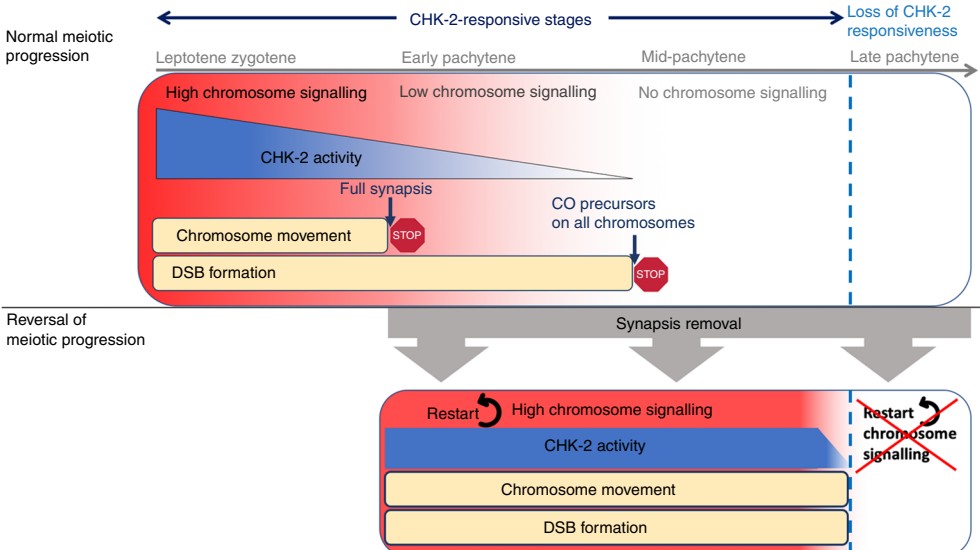

**Fig. 8 Model: chromosome signalling regulates CHK-2 activity to control meiotic progression.** During normal meiotic progression chromosome signalling (red to white gradient) promotes CHK-2 activity (blue to white gradient). The presence of unsynapsed chromosomes during leptotene–zygotene induces high chromosome signalling, which sustains high CHK-2 activity, chromosome movement and DSB formation. Full synapsis causes a reduction of chromosome signalling and CHK-2 activity that terminates chromosome movement (first STOP sign). The formation of crossover precursors on all homologue pairs terminates chromosome signalling, shutting down CHK-2 activity and DSB formation (second STOP sign). Removal of the SC from early and mid-pachytene nuclei, either directly or by cohesin removal, restarts high chromosome signalling, inducing full CHK-2 activity and reactivating chromosome movement and DSB formation. Nuclei that transition to late pachytene lose the capacity to reactivate CHK-2.

Two key aspects of our model for feedback regulation of CHK-2 are that surveillance of the chromosomal features responsible for signal generation must operate continuously until late pachytene, and that signal emission can be reactivated if SC stability is compromised during pachytene. *C. elegans* mutants with defects in synapsis arrest at earlier stages (leptotene/zygotene) than mutants deficient in CO formation but competent in SC assembly, which arrest at early pachytene, suggesting that synapsis and CO formation may be differently monitored to feedback on CHK-2[10,19]. However, recent studies show that the state of the SC is modified by the formation of CO precursors[44–46], opening the possibility that SC state itself may be the feature under surveillance to monitor formation of CO precursors[45]. As such mechanism would operate in a chromosome-autonomous manner, surveillance of SC status can explain how a single homologue pair lacking a CO precursor will keep emitting signals that sustain CHK-2 activity and how such signal will be extinguished once all homologue pairs form CO precursors. We propose that unsynapsed axial elements emit signals that induce/sustain high CHK-2 activity, while synapsed axial elements in the context of homologues lacking a CO precursor emit signals that result in lower CHK-2 activity. Then, recombination-dependent SC stabilisation terminates axis signalling.

Our model requires constant monitoring of SC status between leptotene and late pachytene, which is possible in *C. elegans* because HORMADs remain bound to axial elements throughout pachytene. Mammals also use HORMADs to monitor synapsis[58], but in contrast to *C. elegans* synapsis triggers ejection of HORMADs from axial elements[59], thus terminating axis signalling. Despite these differences in HORMADs behaviour, the role of the SC in coupling the formation of CO precursors to the termination of axis signalling may be conserved between worms and mammals, as SC assembly is recombination dependent in mammals.

Studies in different organisms show that synapsis contributes to implement local and chromosome-wide regulation of events such as DSB and CO formation[60–62]. Our study highlights how SC surveillance contributes to integrate pairing and recombination with meiotic progression by ensuring timely and sustained termination of chromosome movement and DSB formation in a nucleus-wide fashion. This role of the SC must be coordinated with its roles in promoting CO formation locally at CO-designated sites[50], and in limiting DSB and CO formation along paired homologues[44,61]. Thus, signalling coupled to the SC operates in a chromosome-autonomous and nucleus-wide manner to regulate DSB and CO formation, control nuclear organisation, and promote meiotic progression.

## Methods

**C. elegans strains and culture conditions.** All strains were grown on *E. coli* (OP50) seeded NG agar plates at 20 °C under standard conditions. Unless otherwise indicated, all experiments were performed using young adults at 18–24 h post L4. The N2 Bristol strain was used as wild-type strain. The following alleles were used: LG IV: *htp-1 (gk174)*, *rec-8 (ok978)*, *spo-11 (ok79)*; LG V: *coh-3 (gk112)*, *coh-4 (tm1857)*, *syp-2(ok307)*, *chk-2(me64)*. Supplementary Table 1 contains a full list of the strains generated in this study. Strains VC666 [*rec-8 (ok978) / nT1 [unc-? (n754) let-? qls50] (IV;V)*], TY5120 [*coh-3(gk112) coh-4(tm1857) V / nT1 [unc-? (n754) let-? qls50] (IV;V)*], and AV146 [*chk-2(me64) rol-9(sc148)/unc-51(e369) rol-9 (sc148)*] were used for viability assays.

**Generation of transgenic C. elegans strains.** Transgenic strains were generated using CRISPR to modify the endogenous locus or single-copy insertions of the desired transgene in strains carrying the MosCI transposon at the *ttTi5605* (chromosome II), *ttTi4348* (chromosome I), or the *oxTi177* (chromosome IV) loci[63]. The *rec-8^{3XTEV}::GFP* transgene was generated by adding a 75 bp fragment encoding for three repeats of the TEV recognition motif (ENLYFQGA-SENLYFQGELENLYFQG) after REC-8's Q289 codon in a vector expressing REC-8::GFP under the *rec-8* promoter and 3′ UTR[21]. The *coh-3^{3XTEV}::mCherry* transgene was generated by adding a 75 bp fragment encoding for three repeats of the TEV recognition motif (ENLYFQGASENLYFQGELENLYFQG) after COH-3's I315 codon in a vector expressing COH-3::GFP under the *coh-3* promoter and 3′ UTR. The *rec-8::AID::GFP* transgene was generated by adding a 135 bp fragment encoding the 35 amino acids of the AID tag[38] before the start codon of GFP. The *plk-2::GFP* transgene was generated by fusing the endogenous sequence of *plk-2*, including introns, 860 bp of upstream and 1393 bp of downstream sequence to a GFP sequence containing three introns. Supplementary Table 2 contains a list of all transgenes used in this study.

CRISPR-mediated genome editing was performed using preassembled Cas9-sgRNA complexes, single-stranded DNA oligos as repair templates, and *dpy-10* as a co-injection marker as described in[64]. To generate TEV-cleavable versions of

REC-8 and COH-3, a 75 bp fragment encoding for three repeats of the TEV recognition motif (ENLYFQGASENLYFQGELENLYFQG) was inserted after REC-8's Q289 and COH-3's I315 codons using a single-stranded DNA oligo as repair template (Supplementary Table 3). To generate *chk-2::AID* and *syp-2::AID* alleles we introduced 135 bp encoding the 35 amino acids of the AID tag[38] before the stop codon of the genes using two single-stranded DNA oligos with a 35 bp overlap as repair templates (Supplementary Table 3). We also used this strategy to generate an *smc-1::AID::GFP* allele by introducing the 135 bp of the AID tag before the start codon of GFP in the *smc-1* (*fq20[smc-1::GFP]*) allele that we generated previously[21]. Supplementary Table 3 offers a full list of CRISPR alleles generated in this study.

**Auxin-mediated protein degradation**. All strains used for auxin-mediated protein degradation were homozygous for the *ieSi38* transgene expressing the TIR1 protein under the *sun-1* promoter, which confers efficient germline expression[38]. Auxin treatment was performed by placing young adult worms in seeded NG agar plates containing 1 or 4 mM Auxin for the indicated periods of time.

**TEV protease microinjection**. Germline injections were performed in young adult worms at 18–24 h post L4 immobilised in 2% agarose pads and covered with Halocarbon oil 700 (Sigma) using a Narishige IM-31 pneumatic microinjector attached to an inverted Olympus IX71 microscope. Needles were made using borosilicate glass filaments with a 1.0 mm O.D. and 0.58 mm I.D. (BF100-58-10, Sutter Instruments) and a micropipette puller P-97 (Intracell). In all experiments except those displayed in Supplementary Fig. 2a, AcTEV™ Protease (Thermo Fisher, Cat. No. 12575) was used in a mix containing 10U/µl TEV protease in 50 mM Tris-HCl, pH 7.5, 1 mM EDTA, 5 mM DTT, 50% (v/v) glycerol, 0.1% (w/v) Triton X-100. For Supplementary Fig. 2a we used TEV protease from GenScript (Cat. No. Z03030-1000) in a final mix containing 2.5 ng/µl TEV protease in 50 mM Tris, 5 mM DTT, 12.5% glycerol, pH 7.5. In indicated experiments 25 pmol of tetramethyl-rhodamine-5-dUTP (Roche) were added to the injection mix to evaluate the efficiency of germline microinjection by the incorporation of labelled nucleotides into the DNA of germ cells. Following microinjection, worms were rescued from the agarose pad with M9 salt buffer and placed in NG plates with *Escherichia coli* OP50 for 3h30' (unless otherwise indicated) before germline dissection.

**Immunostaining and image acquisition**. Germlines from 18 to 24 h post L4 worms were dissected in EGG buffer (118 mM NaCl, 48 mM KCl₂, 2 mM CaCl₂, 2 mM MgCl₂, 5 mM HEPES) containing 0.1% Tween and then fixed in the same buffer containing 1% paraformaldehyde for 5 min. Slides were immersed in liquid nitrogen before removing the coverslip and then placed in methanol at −20 °C for 5 min. Slides were then washed three times for 10 min each in PBST (1x PBS, 0.1% Tween) and then blocked in PBST containing 0.5% BSA for 1 h. Slides were then incubated with primary antibodies diluted in PBST overnight at room temperature. Following three washes of 10 min each in PBST, slides were incubated in the dark at room temperature for 2 h with secondary antibodies diluted in PBST. All secondary antibodies were conjugated to Alexa-488, Alexa-555 or Alexa-647 (Life Technologies) and used at 1:500. Following three washes with PBST, slides were counterstained with DAPI, washed for 1 h in PBST and mounted using Vectashield (Vector). Unless otherwise indicated, all images were acquired as three-dimensional stacks on a Delta Vision system equipped with an Olympus 1×70 microscope using a ×100 lens. Images were subjected to deconvolution analysis using SoftWoRx 3.0 (Applied Precision) and images were mounted in Photoshop.

**Super resolution structured illumination microscopy**. Immunostaining was performed as described before, but slides were mounted using ProLong Diamond mounting media instead of Vectashield and covered using Zeiss high-performance 0.17 ± 0.005 coverslips. Images were acquired using a Zeiss Elyra S1 SIM microscope, processed with Fiji, and mounted in Photoshop.

**Antibodies used in this study**. The following primary antibodies were used at the indicated dilutions: rabbit anti-GFP-488-conjugated (1:200) (Invitrogen, A21311), goat anti-GFP-FITC-conjugated (1:200) (Abcam, ab6662), rat anti-mCherry (1:1000) (Chromotek, 5F8), rabbit anti-COH-3/4 (1:400)[21], mouse anti-REC-8 (1:100) (Novus Biologicals, 29470002), guinea pig anti-SYP-1 (1:400)[26], chicken anti-SYP-1 (1:300)[47], guinea pig anti-HTP-3 (1:800)[32], rabbit anti-HIM-3 (1:400)[33], rabbit anti-HTP-1/2 (1:400)[47], rabbit anti-PLK-2 (1:500)[65], guinea pig anti-SUN-1 pS12 (1:1000)[19], rabbit anti-RAD-51 (1:10000) (Novus Biologicals, 29480002), mouse anti-HA (1:200) (Cell Signalling, 23675), rabbit anti-DSB-2 (1:1000)[9], rabbit anti-HIM-8 (1:500) (Novus Biologicals, 41980002). Phospho-specific antibodies against HIM-8 pT64 (1:400) were produced by injecting rabbits with the synthetic peptide DTPRFSpTPIVPNVC (GenScript). Polyclonal anti-HIM-8 pT64 antibodies were affinity purified by binding to a column containing the phospho-peptide and the specificity of the antibodies was validated by absence of staining in germlines of *chk-2* mutant worms.

**Quantification of RAD-51 foci**. Analysis of RAD-51 foci was performed as in[47]. Each germline was divided into 6 equal-size regions, from the beginning of the transition zone (leptotene–zygotene) to the bend of the germline (end of pachytene, beginning of diplotene). The number of foci per nucleus in each region of the germline were quantified. When RAD-51 signals appeared as agglomerates instead of single foci they were categorised as stretches. At least three germlines per genotype and condition were used for quantification of RAD-51 foci and the number of nuclei counted for each zone of the germline is shown on Supplementary Table 4.

**Quantification of PLK-2, HIM-8pT64, SUN-1pS12, and DSB-2 staining**. We used the presence of two or more aggregates of PLK-2, HIM-8 pT64, or SUN-1 pS12 on the nuclear envelope as markers of active chromosome movement. To determine the percentage of vertical rows of nuclei displaying these markers we first counted the total number of vertical rows of nuclei between the start of transition zone and the end pachytene. Then, starting from the beginning of transition zone, we identified the last vertical row in which at least 50% of nuclei had two or more aggregates of PLK-2, HIM-8 pT64, or SUN-1 pS12 and counted the number of vertical rows of nuclei included in that section of the germline. The number of vertical rows scored as positive was then normalised to the total number of vertical rows of nuclei in each germline. The same procedure was used to determine the region of DSB-2 positive staining, which we used as a marker of competence for DSB formation. Three germlines per genotype and condition were used for quantification, unless otherwise indicated. In Fig. 7b, PLK-2 aggregates were quantified by dividing the region between meiosis onset and the end of pachytene into six equal-size regions and counting the number of nuclei with 0, 1, or >1 PLK-2 aggregates in each region.

**Western blot**. Whole-worm protein extracts were prepared by picking 150 young adult worms of desired genotype into 1X Laemmli buffer and subjecting them to three cycles of freeze-thawing before boiling the samples for 5 min. Protein extracts were run on 10% acrylamide gels (Bio-Rad) and transferred onto nitrocellulose membrane for 1 h at 4 °C. Following 1 h blocking in 1X TBS 0.1% Tween containing 5% dried milk, the nitrocellulose membranes were incubated overnight with rabbit anti-DSB-2[9] and goat anti-actin (Santa Cruz, SC1616) primary antibodies, used at 1:1000 and 1:3000, respectively, diluted in blocking buffer. After three washes of 10 min each with 1X TBS 0.1% Tween, membranes were incubated for 1 h at room temperature with HRP-conjugated goat anti-rabbit (Jackson Immunoresearch, AB_2307391) (1:5000) and HRP-conjugated donkey anti-goat IgG (Sigma, AP180P) (1:8000) antibodies in blocking buffer. Finally, membranes were washed with 1x TBS 0.1% Tween (three times for 10 min) and treated with ECL™ Western Blotting detection kit (GE Healthcare).

**RNA interference (RNAi)**. All RNAi experiments were carried out by feeding worms with HT115 bacteria transformed with a vector for IPTG-inducible expression of dsRNA. *syp-1* RNAi was performed using clone V-10P20 from the Ahringer library. Bacteria containing the *syp-1* vector, as well as empty vector (HT115) control, were both grown overnight at 37 °C in LB with 50 µg/ml ampicillin. Cultures were then centrifuged at 2095 × *g*, collected and resuspended in 1 ml LB, before seeding 100 µl of this culture onto NGM agar plates containing 1 mM IPTG and 25 µg/ml ampicillin. When RNAi experiments were combined with auxin-mediated protein degradation RNAi plates also contained 1 or 4 mM auxin. Plates were incubated overnight at 37 °C to induce the expression of dsRNA. Eggs from the indicated strains were hatched on RNAi plates and maintained in these plates until they became young adults (18–24 h post L4) and were used for experiments.

**Viability screening**. L4 worms from the indicated strains were individually picked and transferred onto new plates every 12 h, when the total number of embryos on each plate was counted. The presence of dead embryos (embryonic lethality) was assessed 24 h after the mother had been removed from each plate. Five individual worms from each genotype were scored, and the total number of embryos counted is shown (*n*).

**In vivo imaging**. Adult hermaphrodites were immobilised on a slide using a hydrogel-microbead solution similar to previously described protocol[66]. First, adult worms treated for 2 h in OP50 seeded plates containing 1 mM auxin or no auxin controls were placed on a 3 µl droplet containing 0.5 mM levamisole dissolved in S medium to help immobilisation. Next, 7 µl of precooled hydrogel-microbead solution 45% Pluronic F-127 (Sigma ♯P2443) mixed with 30 µm polystyrene microbeads (Sigma ♯84135) were added around the droplet and worms were overlaid with a coverslip containing also 7 µl of precooled hydrogel-microbead solution. Coverslips were sealed using nail polish and imaged immediately in a Delta Vision deconvolution system (Applied precision). 0.8 µm spaced Z-stack images were acquired with a time-lapse of 10 s over 7 min with the following settings: 50% power, 25 ms exposure, ×60 microscope lens and 1024 × 1024 image size.

**Tracking of PLK-2::GFP aggregates movement**. Images from mid-pachytene nuclei were acquired as a series of 1 μm spaced Z-stacks, with a time-lapse of 5 s over 7 min (85 frames), and with the following parameters: 50% light intensity, 0.025 s exposure, ×60 objective and image size of 1024 × 1024 pixels. Maximum intensity projections of the Z-stacks were created using Fiji and the Stackreg Fiji plugin was used to align the stack of images. Imaris version 9.5.1 was used to track PLK-2::GFP aggregates applying the following settings: autoregressive motion algorithm, estimated spot diameter 0.3 μm, maximum distance between spots 0.75 μm, maximum gap size between spots 3. The average speed of each PLK-2::GFP aggregate per nucleus was analysed using Graphpad Prism 8 Nested t test, where columns represent each genotype and subcolumns represent different germlines. The distribution of projected speeds for all PLK-2::GFP aggregates inside a nucleus over 7 min was analysed using Excel.

**RNAseq: RNA extraction**. Four biological replicates were used for each condition. For each replicate, 50 18–24 h post L4 worms were treated for 4 h in OP50 seeded NG plates containing 4 mM auxin or in control plates without auxin. Next, adult worms were collected into 400 μl of M9 salt buffer in RNase-free tubes. After the worms settled to the bottom, the majority of the M9 was removed. 1 ml of Trizol (Thermo Fisher ♯15596026) and 100 μl of 0.5 mm of glass beads (Sigma ♯Z250465) were added to each tube and immediately frozen at −80 °C. Samples were thawed in ice and bead-beaten with a FastPrep-24 5G instrument (MP Biomedicals) with the following settings: speed 4 m/s, 3 cycles, and run time of 20 s. After the beads settled at the bottom of the tube, the solution was transferred to a new RNase-free tube. 200 μl of chloroform was added and tubes were vigorously mixed by hand for 15 s. Samples were then centrifuged at 12,000 g for 15 min at 4 °C and the resulting clear upper-phase was transferred to a new tube, where 500 μl of isopropanol and 1 μl of glycogen were added prior to storing the sample overnight at −20 °C. Next, samples were spun at 12,000 g for 10 min at 4 °C, the supernatant was discarded and the pellet washed with 1 ml of fresh 75% ethanol. After 5 min centrifugation at 7500 g and 4 °C, ethanol was discarded, pellet briefly air-dried and resuspended with 15 μl of RNase-free water.

**RNAseq: sequencing and analysis**. Paired end 100 bp libraries were sequenced on Illumina Hiseq 2500 and Raw basecall files were converted to fastq files using Illumina's bcl2fastq (version 2.1.7). Reads were aligned to the *C. elegans* genome (ce10) using Tophat2 version 2.0.11[67] with default parameters. Mapped reads that fell on genes were counted using featureCounts from Rsubread package[68] version 1.20.6. Generated count data were then used to identify differentially expressed genes using DESeq2[69] version 1.16.1. Genes with very low read counts were excluded.

RNAseq data set GEO accession number: GSE134989 (https://www.ncbi.nlm.nih.gov/geo/query/acc.cgi?acc=GSE134989)

**Reporting summary**. Further information on research design is available in the Nature Research Reporting Summary linked to this article.

## Data availability

RNAseq data set for Fig. 5a can be accessed using GEO accession GSE134989. Source data are provided with this paper. All other relevant data are available within the Article and Supplementary Information files or available from the authors upon reasonable request. *C. elegans* strains generated in this study are available from the corresponding author.

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

## Acknowledgements

We thank Laurence Game and Ivan Andrew from the MRC LMS Genomics facility for library preparation for RNAseq experiments, the *Caenorhabditis* Genetics Center (CGC) for providing *C. elegans* strains, and the following individuals for providing antibodies: Anne Villeneuve, Monique Zetka, Verena Jantsch, and Rueyling Lin. This work was supported by an MRC core-funded grant to E.M.-P. and by postdoctoral Fellowships from Fundación Alfonso Martín Escudero and EMBO to M.C.-P.

## Author contributions

Conceptualisation E.M.P., M.C.-P., and S.P.; methodology M.C.-P. and S.P. Investigation M.C.-P., S.P., G.S., A.L.J.-S., M.H.D., M.M.K.; writing E.M.-P., M.C.-P., and S.P. Funding acquisition E.M.P. and M.C.-P.

## Competing interests
The authors declare no competing interests.
