## [Peer Review File · Nature Communications]

Reviewers' comments:

Reviewer #1 (Remarks to the Author):

This is an important manuscript by Castellano-Pozo et al., investigating the requirements for meiotic checkpoint satisfaction in *C. elegans*. By acutely disrupting the integrity of the meiotic axis or synaptonemal complex components directly, the authors destabilize the synaptonemal complex and reactivate early events in meiotic prophase, namely chromosome mobility and competence for DSB formation. They show that this reactivation relies on continuous CHK-2 activity, HTP-1 function, the integrity of the meiotic axis (or at least the presence of meiotic HORMADs on meiotic axes) and is limited to nuclei in early and mid-pachytene. The experiments are well done and the conclusions contribute dramatically to our understanding of the regulation of meiotic prophase events.

I have a few major concerns:

Considering the role that chromosome mobility is thought to play in mediating homolog pairing, what effect does asynapsis and reactivation of CHK-2 dependent events have on meiotic pairing? Is homolog pairing lost at PC and/or non-PC sites? I'd suggest performing this experiment in both situations where synapsis is destabilized directly (by disrupting SC components) and indirectly (by disrupting cohesin), since this might produce different results.

I appreciated the inclusion of movies depicting the increased mobility of chromosomes when synapsis is destabilized. Can the authors provide quantification of this movement to see if it accurately recapitulates movement seen when synapsis is completely disrupted, for example, in *syp-1* mutants?

Perhaps I missed it but can the authors provide viability analysis of the following strains:
rec-8(3XTEV);coh-3(3XTEV); coh-4
chk-2::AID

Pg 12: "Homozygous *chk-2::AID* worms (generated by CRISPR) displayed normal chiasma formation despite slower SC assembly." Is this data included somewhere or is it data not shown?

Minor

There are examples where there are misspellings/typos or the language could be made more clear or accurate:

pg 4: "In wild-type germlines synapsis induces termination of CHK-2-*dependent* chromosome movements by pachytene entrance"

pg 14: "Therefore, the striking changes in nuclear organization caused by SC disassembly occur in the absence of obvious transcriptional changes *of proteins known to be involved in meiosis*"

Reviewer #2 (Remarks to the Author):

Overview

Here, the authors present a well-constructed investigation into how meiotic chromosome structure—mediated via the synaptonemal complex (SC) and cohesin complexes—influences early stages of recombination and inter-homologue chromosome interactions. In recent years, it has been recognised that there is an interplay between chromosome synapsis (tight coalignment by the SC) and early prophase events (e.g. in mouse, <https://www.ncbi.nlm.nih.gov/pubmed/19851446>; in budding yeast, <https://www.ncbi.nlm.nih.gov/pubmed/24717437>). This study builds on these concepts in *C.*

elegans using inactivation of key components, and uncovers a direct role for the regulation of CHK2 kinase in this process.

General comment: I would strongly urge the authors to alter their future submissions (and revisions) in three simple ways that would greatly enhance the reviewing experience:

1. Please prepare figures and legends on the same page so that the figure can be directly read without having to leaf back and forward through the document. The more accessible is the submission, the more likely it is to get both a constructive and speedy review.
2. Please provide line numbering so that reviewer comments can directly reference relevant text in a simple manner.
3. Please consider splitting up large paragraphs to make things easier for the reader.

Specific critique follows below:

Page 6. COH3 and COH4 are introduced as redundant players, but from that point on, only COH3 is explored. It would help if the authors could clarify the degree of redundancy, and what complexes/structures are expected to be retained upon depletion of only COH3. Might this affect any interpretations?

Page 6-7. It would be helpful to explicitly note that the temporal organisation of the worm germline enables temporal exploration of function without needing a direct live-cell imaging approach.

Page 7, bottom, reference to Fig 2B. "Following...". There are two statements here, neither has quantification in the figure ("...each...was associated...", and "...most nuclei..."). Please provide evidence for both. For example, colocalisation in Fig 2B is not obvious to this reviewer.

Page 8. The auxin experiments seem to require treatment for 8–16 hours. This seems like it might muddy the ability to ascribe a temporal stage to the effect. It help greatly would help if this were commented upon and/or clarified.

From an important mechanistic point, if the depletion/phenotype takes a long time, one can imagine that levels of active CHK2 are slowly changing. Since part of their model (Fig 7C) appears to involve differential effects depending on the relative amount of CHK2 activity, might there not be odd effects occurring? Please can this be commented and, if necessary, be addressed?

Page 9..., and quantification in figures 3-4. The authors use the quantity: "% of rows of nuclei with...". Please can the rationale for this be explained? Are the rows vertical or horizontal relative to the germline temporal order? Why is the quantity % of rows and not simply % of total nuclei? These analyses require more careful explanation before I feel confident that I can or cannot agree with the conclusions. Moreover, how many worms/germlines in total are analysed?

Page 14. The authors conclude that the reactivation of CHK2 occurs without changing to CHK2 transcription. However, this experiment was only performed in the SYP2 depletion system, yet a general effect is inferred. The authors should either also include transcriptomics in the cohesin depletion system, or should clarify that they assume/infer the same to be happening upon ectopic loss of REC8 or COH3.

Page 15. Finally, the observations throughout the paper that late-stage pachytene nuclei appear resistant to the effects of auxin or cohesin loss are interesting and potentially highly revealing. It would help if these could be put into broader context to help understand the mechanistic basis of what has been presented. For example, if I understood the study correctly, it is inferred that the regulation of CHK2 (in response to synapsis status) is mediated via HTP1. Therefore it would seem important to explore and/or comment upon the known association of HORMA proteins at late

pachytene. Are they still present? Or some factors lost from chromosomes (or permanently inactivated) at this stage?

Discussion. General comment: It feels rather long, has a number of very long paragraphs that are unwieldy to parse, and has only a limited number of section headers covering a large amount of text. If possible, I suggest trimming down the content, and splitting up the text with more paragraph breaks and an increased number of shorter headings so that clear conclusions and discussion points can be made more accessible.

Reviewer #3 (Remarks to the Author):

In the current work, the authors combine the experimental advantages of the worm germline with novel temporally-resolved protein removal methods to understand how cohesin and subsequently the central element proteins of the SC contribute to meiotic chromosome function. The paper is well designed experimentally and the results are novel and most are unexpected in the field providing new insights into how the central element region of the SC is an essential player that integrates pairing and recombination with early meiotic progression in a predictable CHK-2 dependent manner. The observation of the involvement of REC8 and or COH3/4 in the stabilization of the central element of the SC (synapsis) is a novel result and the starting point that the authors use to show how the disassembly of the central element of the SC give rise to the unpredicted and unexpected reversibility of the meiotic progression. This central-element-dependent reversibility of the meiotic program is a novel result of great relevance in biology. Following with the argument, the authors further show that this process is mechanistically dependent on CHK-2 and HORMAD. Altogether, these results provide novel and very interesting insight of how the nucleus-wide loss in competence for DSB formation and repair is regulated in a model organism like the worm and opens up the very likely possibility that a similar mechanism would be operating in mouse and humans.

The results are thus of great relevance and are supported by well performed and elegant experiments making use of novel genetic tools. The MS (excluding the abstract, see below) is very well written and the results are shown orderly and explained in a clarity and hypothesis driven way.

Main comments to the experiments carried out:

1-The authors generate and functionally validate kleisin alleles that can be proteolitically controlled by introducing TEV motifs in REC8-GHP and COH-3-Cherry. This is shown in figure 1 by IF. The results are quite robust and clearly show how a single injection of the TEV protease can proteolitically remove the REC8 and or COH3 kleisins.

2-In relation to figure 2, when comparing the results with those of figure 1 I can not see the absence of the aggregates observed in figure2 at figure 1, that is the appearance of aggregates of REC8 at COs (positive for COSA1). One would expect similar results given that the genotypes and treatment are apparently similar.

3-The results showing a similar phenotype removing either REC8 or COH3 or both simultaneously is of interest. This redundancy should be addressed in the text given the supposed specific functions that each of the kleisin play in meiosis.

4-The results showing how the meiotic kleisins are required for central element stability (SC) are surprising but especially striking is the observation that cohesins are not able to be released around the COs after TEV treatment. How is this proteolysis prevented??? How are the meiotic cohesins stabilized at the COs? Or are they resistant to proteolysis?

5-The results showing that both cohesins promote axis integrity are as expected and were validated in part with the SMC1-degron system as shown in figure S2D-E and figure 2E. This result is of interest and validated the former result. I would suggest to carry out a localization of COSA1 in the SMC1::AID::GFP (16 h auxin) to demonstrate and conclude that the accumulation of cohesins at COs is dependent on whole cohesins (SMC1-containing complexes).

6- Next the authors, based on the observation of chromatin shape, show for the first time that chromosome clustering is regained in pachytene cells that have been removed of cohesins Rec8 or COH-3 by TEV protease treatment. These results suggest the regaining of chromosome movement in pachytene. This was clearly shown indirectly in Figure 3 by staining with PLK2 and its downstream substrate SUN1-S12p. It should be interesting to repeat these experiments for robustness.

7-Following the "de-differentiation" after cohesin removal, the authors convincingly show in figure 3E DSB-2 and RAD51 staining in a SPO11-dependent manner. These results strongly suggest intuitively that upon cohesin depletion meiosis is to some extent "reinitiated". However, the explanation of the increase of RAD51 foci given by the authors is not clear and points toward two directions (unrepaired + de novo DSBs) in a very equidistant manner. This point could be further clarified.

8-The authors next address if central element disassembly as a consequence of cohesin removal is directly causing CHK2 reactivation. To do that, the authors make use of the auxin-inducible system at the SYP-2 loci. The results are consistent with all the meiotic reentry phenotype as shown in figure 4, though to a lesser extent (40% DSB-2 at figure 3C-D vs 50% of DSB-2 at figure 4C). This would need some explanation.

9-Next and following a logical rationale, the authors try to validate a CHK-2 dependent meiotic re-entry upon SC depletion. The CHK-2-auxin system developed shows elegantly loss of PLK2 function after 6 hours of auxin treatment validating the model. As expected, SC disassembly (REC8-TEV) in the absence of CHK-2 failed to induce PLK2-aggregates (Figure 5). In order to get a more robust conclusion, recombination markers should also be evaluated under these circumstances.

10-Next, and in order to determine if SC-disassembly-dependent re-entry into early meiosis is associated to transcriptional activity, the authors carry out RNAseq and conclude that, as expected from a CHK-2-dependent pathway, it is independent of new transcription. The western blot analysis of DSB-2 showing similar protein levels could also be carried out with an additional marker previously used (whenever technically possible) to demonstrate that protein levels are unchanged after depletion of cohesins REC8 and COH3.

11-In order to validate the conclusion that Cohesin-containing axial elements contribute to maintain CHK-2 activation (Figure 7), the SMC1 model that the authors have used previously in this work would validate it conclusively better than with the double COH3 and REC8.

Main formal comments to the MS

1-From my point the most controversial part of the MS is the title and the abstract. The title from my personal perspective could be drastically improved.

Regarding the abstract, it should be re-written for a more general reader. I would suggest to mention "the removal of the central element of the SC" instead of "removal of the SC" as the responsible of the phenotype observed. The SC is well referred to the tripartite structure but sometimes the axial elements or chromosome axes are also referred (non-orthodox way) as the

growing SC. It should also be emphasized in the abstract the reversibility (re-entry into early prophase I) of the meiotic programme when the central element integrity is altered with different mutants (CE and cohesin axes). In general, the abstract of the paper does not summarize the experimental results that lead to the conclusions but mostly it is a list of conclusions in an abstracted way. I would suggest to summarize the experimental results that lead to a pair of clear general conclusions.

We thank all three reviewers for their overall support and constructive criticism of our manuscript. Below we provide detailed answers to all reviewer's comments.

Reviewer #1 (Remarks to the Author):

This is an important manuscript by Castellano-Pozo et al., investigating the requirements for meiotic checkpoint satisfaction in *C. elegans*. By acutely disrupting the integrity of the meiotic axis or synaptonemal complex components directly, the authors destabilize the synaptonemal complex and reactivate early events in meiotic prophase, namely chromosome mobility and competence for DSB formation. They show that this reactivation relies on continuous CHK-2 activity, HTP-1 function, the integrity of the meiotic axis (or at least the presence of meiotic HORMADs on meiotic axes) and is limited to nuclei in early and mid-pachytene. The experiments are well done and the conclusions contribute dramatically to our understanding of the regulation of meiotic prophase events.

I have a few major concerns:

Considering the role that chromosome mobility is thought to play in mediating homolog pairing, what effect does asynapsis and reactivation of CHK-2 dependent events have on meiotic pairing? Is homolog pairing lost at PC and/or non-PC sites? I'd suggest performing this experiment in both situations where synapsis is destabilized directly (by disrupting SC components) and indirectly (by disrupting cohesin), since this might produce different results.

We have monitored pairing of the X chromosomes following direct SC destabilisation with the *syp-2::AID*, as well as indirectly by removing cohesin using the *coh-3^{3XTEV}::mCherry; coh-3Δ; coh-4Δ* strain. In both cases we observe that the pairing-center end of the X chromosomes (monitored using anti-HIM-8 antibodies) is fully paired in mid pachytene nuclei following SC disassembly. These results provide further support for our proposal that SC disassembly from pachytene nuclei induces CHK-2-dependent chromosome movements that promote homologue pairing. These results are now mentioned in page 11 lines 24-25 and Figures S4 B-C show quantification of X chromosome pairing following SYP-2 depletion and COH-3 removal, as well as examples of these experiments.

I appreciated the inclusion of movies depicting the increased mobility of chromosomes when synapsis is destabilized. Can the authors provide quantification of this movement to see if it accurately recapitulates movement seen when synapsis is completely disrupted, for example, in *syp-1* mutants?

We have performed additional in vivo imaging experiments to quantify the movement of PLK-2 aggregates following SC disassembly from pachytene nuclei using the *syp-2::AID* after just 2 hours of auxin treatment, and also in strains carrying a null allele of *syp-2* and the *plk-2::GFP* transgene. Tracking and quantification of the movement of PLK-2::GFP aggregates in both strains confirms that removing the SC from pachytene nuclei causes reacquisition of chromosome movements. Importantly, the movement of PLK-2::GFP aggregates in *syp-2::AID* nuclei after two hours of auxin treatment is not significantly different from that observed in mid pachytene nuclei of *syp-2* mutants. These data is now mentioned in page 11 lines 22-23 and shown as a new panel (D) in Figure 4.

Perhaps I missed it but can the authors provide viability analysis of the following strains:
rec-8(3XTEV);coh-3(3XTEV); coh-4
chk-2::AID

These data is now added to the viability Table shown in panel A of Figure S1.

Pg 12: "Homozygous *chk-2::AID* worms (generated by CRISPR) displayed normal chiasma formation despite slower SC assembly." Is this data included somewhere or is it data not shown?

Figure S6A now shows quantification of chiasma formation in diakinesis nuclei of untreated *chk-2::AID* worms, confirming normal chiasma formation, while Figure S6B shows delayed SC assembly in untreated *chk-2::AID* worms compared to WT controls.

Minor

There are examples where there are misspellings/typos or the language could be made more clear or accurate:
pg 4: "In wild-type germlines synapsis induces termination of CHK-2-*dependent* chromosome movements by pachytene entrance"

pg 14: "Therefore, the striking changes in nuclear organization caused by SC disassembly occur in the absence of obvious transcriptional changes *of proteins known to be involved in meiosis*"

We thank the reviewer for spotting these mistakes, we have made both corrections.

Reviewer #2 (Remarks to the Author):

Overview

Here, the authors present a well-constructed investigation into how meiotic chromosome structure—mediated via the synaptonemal complex (SC) and cohesin complexes—influences early stages of recombination and inter-homologue chromosome interactions. In recent years, it has been recognised that there is an interplay between chromosome synapsis (tight coalignment by the SC) and early prophase events (e.g. in mouse, <https://www.ncbi.nlm.nih.gov/pubmed/19851446>; in budding yeast, <https://www.ncbi.nlm.nih.gov/pubmed/24717437>). This study builds on these concepts in *C. elegans* using inactivation of key components, and uncovers a direct role for the regulation of CHK2 kinase in this process.

General comment: I would strongly urge the authors to alter their future submissions (and revisions) in three simple ways that would greatly enhance the reviewing experience:

1. Please prepare figures and legends on the same page so that the figure can be directly read without having to leaf back and forward through the document. The more accessible is the submission, the more likely it is to get both a constructive and speedy review.
2. Please provide line numbering so that reviewer comments can directly reference relevant text in a simple manner.
3. Please consider splitting up large paragraphs to make things easier for the reader.

We have made the format and style changes suggested by the reviewer.

Specific critique follows below:

Page 6. COH3 and COH4 are introduced as redundant players, but from that point on, only COH3 is explored. It would help if the authors could clarify the degree of redundancy, and what complexes/structures are expected to be retained upon depletion of only COH3. Might this affect any interpretations?

The functional redundancy of COH-3 and COH-4 was established by the Meyer group by showing that while *coh-3* and *coh-4* single mutants produced normal levels of viable progeny, *coh-3 coh-4* double mutants displayed high embryonic lethality and severe meiotic defects (Severson et al. 2009). We now cite this study in the opening results section introducing REC-8 and COH-3/4 (Page 6 line 6). In addition, we have performed the experiment requested by the reviewer by introducing the *coh-3^{3XTEV}::mCherry* transgene in worms homozygous for a *coh-3* null allele, but carrying the wild-type allele of *coh-4*. TEV-mediated removal of COH-3^{3XTEV}::mCherry from pachytene nuclei in this background failed to induce SC disassembly, in contrast to the SC disassembly observed when COH-3^{3XTEV}::mCherry is removed from worms of *coh-3 coh-4* background. This result is now mentioned in page 7 lines 22-24 and shown in Figure S2B. We also investigated the effect of COH-3^{3XTEV}::mCherry removal in axial elements containing endogenous COH-4. As expected, we observed that HTP-3 remained associated with axial elements. This is now mentioned in page 8 lines 14-15 and shown in Figure S2G. These results are consistent with COH-3 and COH-4 contributing redundantly to SC stability in pachytene nuclei.

Page 6-7. It would be helpful to explicitly note that the temporal organisation of the worm germline enables temporal exploration of function without needing a direct live-cell imaging approach.

We thank that reviewer for this suggestion. We now make it clear that the temporal organization of the *C. elegans* germline combined with microinjection of the TEV protease into the germline allowed us to directly test the functional requirement of REC-8 and COH-3/4 cohesin at all substages of pachytene (page 7 lines 6-9). We have also modified Figure 1A by substituting the germline diagram by an actual image of a dissected germline stained with DAPI in which we have highlighted nuclei undergoing progression from leptotene to late pachytene.

Page 7, bottom, reference to Fig 2B. "Following...". There are two statements here, neither has quantification in the figure ("...each...was associated...", and "...most nuclei..."). Please provide evidence for both. For example, colocalisation in Fig 2B is not obvious to this reviewer.

We now provide quantification of COSA-1 foci in late pachytene following TEV-mediated removal of REC-8, confirming that most (73% of nuclei) display 6 COSA-1 foci (Figure S2C). We also provide a new figure (Figure S2D) showing an enlarged image of late pachytene nuclei from *rec-8^{3XTEV}::GFP rec-8Δ; cosa-1::HA* worms 3.5 hours after TEV injection showing the overlap of COSA-1 foci with short stretches of REC-8 using arrowheads. This figure also includes quantification of the overlap of REC-8 stretches with COSA-1 foci, showing that 92% of REC-8 stretches in late pachytene nuclei colocalise with a COSA-1 focus. These new quantifications are referred to in page 8 lines 2-4.

Page 8. The auxin experiments seem to require treatment for 8–16 hours. This seems like it might muddy the ability to ascribe a temporal stage to the effect. It help greatly would help if this were commented upon and/or clarified.

The reviewer is correct that the auxin degron approach is slower than the TEV system to remove cohesin from pachytene chromosomes. We clearly state this on page 8 lines 26-27. These experiments were performed to validate some of our observations with the TEV system, in particular the contribution of both REC-8 and COH-3/4 cohesin to axis stability in pachytene nuclei. We confirmed this by depleting SMC-1 (a common component of REC-8 and COH-3/4 cohesin) for 8 hours, which induced local loss of both cohesin and axial elements in mid pachytene nuclei (Figures S2H-I). We also show full disassembly of axial elements in mid-late pachytene nuclei after 16 hours of auxin treatment (Figure 2E). Importantly, since nuclei take about 36 hours to progress through pachytene moving at a rate of 1 row per hour (see new Figure 1A and legend), nuclei that are in the mid-late pachytene region of the germline after 16 hour of auxin treatment had entered the pachytene stage before worms were exposed to auxin. We now clarify this point in lines 30-32 of page 8.

From an important mechanistic point, if the depletion/phenotype takes a long time, one can imagine that levels of active CHK2 are slowly changing. Since part of their model (Fig 7C) appears to involve differential effects depending on the relative amount of CHK2 activity, might there not be odd effects occurring? Please can this be commented and, if necessary, be addressed?

We agree with the reviewer that in experiments involving longer depletion times (8-16 hours) of cohesin the levels of CHK-2 activity may change over time. However, our main conclusions regarding how the stability of cohesin-containing axial elements and the SC contribute to regulate CHK-2 activity are derived from the TEV-mediated depletion of cohesin in 3.5 hours and from the auxin-mediated depletion of the SC in 4 hours. Moreover, we also show reactivation of chromosome movement in mid pachytene nuclei just 1.5 hours after TEV-mediated depletion of REC-8 (Figure S5A) and reactivation of chromosome movement following just 2 hours of direct SC disassembly using the *syp-2* degron system (Figure 4C and Movies S1-4), which is now reinforced by new in vivo imaging experiments following direct SC disassembly (Figure 4D and Movies S5-6). All these experiments clearly show rapid reimplementation of CHK-2-dependent events in pachytene nuclei, including reactivation of chromosome movement, which as depicted in our model requires high levels of CHK-2 activity.

Page 9..., and quantification in figures 3-4. The authors use the quantity: "% of rows of nuclei with...". Please can the rationale for this be explained? Are the rows vertical or horizontal relative to the germline temporal order? Why is the quantity % of rows and not simply % of total nuclei? These analyses require more careful explanation before I feel confident that I can or cannot agree with the conclusions. Moreover, how many worms/germlines in total are analysed?

We have revised the main text, figures, and figure legends to clarify how we performed quantifications of nuclei displaying CHK-2-dependent markers in the germline. As indicated in the new Figure 1A, the germline can be divided into vertical rows of nuclei progressing from leptotene to late pachytene, with each germline typically containing around 40 such vertical rows. The word "vertical" was missing in our previous explanation of the method, which made our explanation difficult to follow, as highlighted by the reviewer. We now clearly explain this the first time that the quantification method is mentioned (page 9 lines 28-31) and also direct the reader to Figure 1A as a reminder of how the germline can be divided into vertical rows of nuclei. Once the germline is divided into vertical rows we identify the last row in which more than 50% of the nuclei are positive for the marker under investigation and use that to determine the % of vertical rows positive for that marker. A detailed description of the method used for quantification is offered in the "Quantification of PLK-2, HIM-8 pT64, SUN-1 pS12, and DSB-2 staining" section of Methods described in supplemental information. This method provides an easy way of visualising the extent of meiotic prophase nuclei displaying specific markers. We provide multiple examples of full-length germlines (Figures S3, S4, S5, S7, and S8) where the extent of nuclei positive or negative for different CHK-2-dependent markers can be easily visualised.

Three germlines (each containing around 300 nuclei between leptotene and late pachytene distributed in around 40 vertical rows) were quantified for most experiments. The number of germlines included in each graph is mentioned in the corresponding figure legend. Furthermore, we have modified all graphs showing whole-germline quantifications to include each individual data point and we also provide the raw values for all data shown in graphs in the Source Data file.

Page 14. The authors conclude that the reactivation of CHK2 occurs without changing to CHK2 transcription. However, this experiment was only performed in the SYP2 depletion system, yet a general effect is inferred. The authors should either also include transcriptomics in the cohesin depletion system, or should clarify that they assume/infer the same to be happening upon ectopic loss of REC8 or COH3.

Performing transcriptomics analysis following TEV-mediated removal of COH-3 or REC-8 cohesin is not technically feasible. First, the TEV protease is only injected into one of the two germlines present in each worm as injecting both germlines is technically challenging and increases lethality of injected worms due to mechanical damage. As whole worms are used for RNA extraction, worms in which only one of the two germlines is injected are not usable for RNA seq experiments. Second, the number of worms required for transcriptomics analysis exceeds what can be realistically achieved by microinjecting individual worms, which is required for the TEV-mediated protein depletion. Therefore, we now make it clear that, based in our transcriptomics experiments using the *syp-2* degron, we infer that the nuclear reorganization observed when REC-8 or COH-3 cohesin are removed using the TEV system also takes place without transcriptional changes of the meiotic machinery (page 15 lines 10-13).

Page 15. Finally, the observations throughout the paper that late-stage pachytene nuclei appear resistant to the effects of auxin or cohesin loss are interesting and potentially highly revealing. It would help if these could be put into broader context to help understand the mechanistic basis of what has been presented. For example, if I understood the study correctly, it is inferred that the regulation of CHK2 (in response to synapsis status) is mediated via HTP1. Therefore it would seem important to explore and/or comment upon the known association of HORMA proteins at late pachytene. Are they still present? Or some factors lost from chromosomes (or permanently inactivated) at this stage?

We thank the reviewer for this suggestion. We have incorporated a sentence in the discussion addressing changes to chromosome structure in late pachytene may be linked the regulation of CHK-2 activity (page 18 lines 11-14).

Discussion. General comment: It feels rather long, has a number of very long paragraphs that are unwieldy to parse, and has only a limited number of section headers covering a large amount of text. If possible, I suggest trimming down the content, and splitting up the text with more paragraph breaks and an increased number of shorter headings so that clear conclusions and discussion points can be made more accessible.

We have shortened the discussion. We have removed headings of sections within the discussion in accordance with journal style.

Reviewer #3 (Remarks to the Author):

In the current work, the authors combine the experimental advantages of the worm germline with novel temporally-resolved protein removal methods to understand how cohesin and subsequently the central element proteins of the SC contribute to meiotic chromosome function. The paper is well designed experimentally and the results are novel and most are unexpected in the field providing new insights into how the central element region of the SC is an essential player that integrates pairing and recombination with early meiotic progression in a predictable CHK-2 dependent manner. The observation of the involvement of REC8 and/or COH3/4 in the stabilization of the central element of the SC (synapsis) is a novel result and the starting point that the authors use to show how the disassembly of the central element of the SC give rise to the unpredicted and unexpected reversibility of the meiotic progression. This central-element-dependent reversibility of the meiotic program is a novel result of great relevance in biology. Following with the argument, the authors further show that this process is mechanistically dependent on CHK-2 and HORMAD. Altogether, these results provide novel and very interesting insight of how the nucleus-wide loss in competence for DSB formation and repair is regulated in a model organism like the worm and opens up the very likely possibility that a similar mechanism would be operating in mouse and humans.

The results are thus of great relevance and are supported by well performed and elegant experiments making use of novel genetic tools. The MS (excluding the abstract, see below) is very well written and the results are shown orderly and explained in a clarity and hypothesis driven way.

Main comments to the experiments carried out:

1-The authors generate and functionally validate kleisin alleles that can be proteolitically controlled by introducing TEV motifs in REC8-GHP and COH-3-Cherry. This is shown in figure 1 by IF. The results are quite robust and clearly show how a single injection of the TEV protease can proteolitically remove the REC8 and or COH3 kleisins.

We thank the reviewer for the supportive comments on our experimental system.

2-In relation to figure 2, when comparing the results with those of figure 1 I can not see the absence of the aggregates observed in figure2 at figure 1, that is the appearance of aggregates of REC8 at COs (positive for COSA1). One would expect similar results given that the genotypes and treatment are apparently similar.

Following TEV-mediated removal of REC-8, persistent REC-8 signals associated with crossover sites are only visible in late pachytene nuclei, when 6 COSA-1 foci emerge in wild-type germlines, as shown in Figure 2B and now also in Figure S2D, which includes a quantification that confirms colocalization of persistent REC-8 signals with COSA-1 foci. Figure 1B shows transition zone and mid-pachytene nuclei, which typically retain 1-2 (some time none) REC-8 signals. Mid-pachytene nuclei shown in Figure 2A also show nuclei with 1-2 REC-8 signals following TEV injection. Figure S1D shows a whole germline 3.5 hours after TEV injection in which the transition from nuclei containing 1-2 REC-8 signals in early and mid pachytene to nuclei containing ~6 signals in late pachytene can be observed.

3-The results showing a similar phenotype removing either REC8 or COH3 or both simultaneously is of interest. This redundancy should be addressed in the text given the supposed specific functions that each of the kleisin play in meiosis.

We have added a sentence at the start of the discussion to highlight how our study reveals novel, redundant and non-redundant, roles for REC-8 and COH-3/4 cohesin in pachytene nuclei (page 17 lines 8-13).

4-The results showing how the meiotic kleisins are required for central element stability (SC) are surprising but especially striking is the observation that cohesins are not able to be released around the COs after TEV treatment. How is this proteolysis prevented??? How are the meiotic cohesins stabilized at the COs? Or are they resistant to proteolysis?

We have also highlighted the local protection of cohesin around CO sites on the first paragraph of the discussion (page 17 lines 13-15). Importantly, we now also show that cohesin is locally protected around CO sites also when SMC-1 is depleted using our SMC-1 degron system (see response to point 5 below and Figure S2J). Therefore, the protection that we observed previously with the TEV system seems to reveal an intrinsic property of cohesin complexes located in the vicinity of CO sites. A recent study (Woglar and Villeneuve, 2018) showed that the structure of the central region of the SC is locally modified at CO sites, our findings now suggest that cohesin complexes around CO sites are also modified in a manner that makes them more resistant to removal by two completely different methods.

5-The results showing that both cohesins promote axis integrity are as expected and were validated in part with the SMC1-degron system as shown in figure S2D-E and figure 2E. This result is of interest and validated the former result. I would suggest to carry out a localization of COSA1 in the SMC1::AID::GFP (16 h auxin) to demonstrate and conclude that the accumulation of cohesins at COs is dependent on whole cohesins (SMC1-containing complexes).

We have performed the experiment requested by the reviewer, which confirms the local protection of cohesin around CO sites. Figure S2J shows that auxin-mediated depletion of SMC-1 for 16 hours results in the persistence of short SMC-1 tracks in late pachytene nuclei and that, crucially, these tracks colocalise with COSA-1 foci. This result is described in page 9 lines 1-3 and we have also included a new paragraph in the discussion to highlight the protection of cohesin around CO sites (see response to point 4 above).

6- Next the authors, based on the observation of chromatin shape, show for the first time that chromosome clustering is regained in pachytene cells that have been removed of cohesins Rec8 or COH-3 by TEV protease treatment. These results suggest the regaining of chromosome movement in pachytene. This was clearly shown

indirectly in Figure 3 by staining with PLK2 and its downstream substrate SUN1-S12p. It should be interesting to repeat these experiments for robustness.

We have repeated this experiment by introducing 3 copies of the TEV cleavage motif using CRISPR into the endogenous *rec8* locus. Figure S3B now shows that similar to the transgene-expressed REC-8^{3XTEV::GFP} used in most experiments of the manuscript, TEV-mediated removal of endogenous REC-8 induces reappearance of chromosome clustering and formation of PLK-2 aggregates on the nuclear envelope of pachytene nuclei, thus fully validating our previous observations.

7-Following the “de-differentiation” after cohesin removal, the authors convincingly show in figure 3E DSB-2 and RAD51 staining in a SPO11-dependent manner. These results strongly suggest intuitively that upon cohesin depletion meiosis is to some extent “reinitiated”. However, the explanation of the increase of RAD51 foci given by the authors is not clear and points toward two directions (unrepaired + de novo DSBs) in a very equidistant manner. This point could be further clarified.

We fully agree with the reviewer that our results are consistent with *de novo* DSB formation following TEV-mediated removal of cohesin or auxin-mediated depletion of the SC. We have rephrased the sentence stating this conclusion to make it clear that our results suggest that REC-8 removal induces *de novo* DSB formation in pachytene nuclei (page 10 lines 25-27).

8-The authors next address if central element disassembly as a consequence of cohesin removal is directly causing CHK2 reactivation. To do that, the authors make use of the auxin-inducible system at the SYP-2 loci. The results are consistent with all the meiotic reentry phenotype as shown in figure 4, though to a lesser extent (40% DSB-2 at figure 3C-D vs 50% of DSB-2 at figure 4C). This would need some explanation.

The overall extension of nuclei positive for DSB-2 staining is very similar following REC-8 or COH-3 removal using the TEV system (Figures 3C-D) and SYP-2 depletion using the degron system (Figure S4D), reaching around 80% in all cases. The reviewer is correct that the base line of nuclei positive for DSB-2 is slightly higher in germlines of untreated *syp-2::AID* worms than in germlines of non-injected *rec-8^{3XTEV::GFP}* or *COH-3^{3XTEV::mCherry}* worms, probably reflecting slight leakiness of the auxin degron system. Regardless of this, our results clearly demonstrate that auxin treatment of *syp-2::AID* worms leads to rapid and complete SC disassembly and to the regaining of chromosome clustering, PLK-2 aggregates on the nuclear envelope, DSB-2 localization to chromosomes, and increased RAD-51 foci. These observations demonstrate the rapid reactivation of early prophase, CHK-2-dependent events, in mid pachytene nuclei following SC disassembly. Moreover, our new in vivo imaging of PLK-2::GFP aggregates in pachytene nuclei of *syp-2::AID* worms treated with auxin for just two hours are fully consistent with the reimplementation of functional chromosome movement in pachytene nuclei (Figure 4D and Movies S5-6).

9-Next and following a logical rationale, the authors try to validate a CHK-2 dependent meiotic re-entry upon SC depletion. The CHK-2-auxin system developed shows elegantly loss of PLK2 function after 6 hours of auxin treatment validating the model. As expected, SC disassembly (REC8-TEV) in the absence of CHK-2 failed to induce PLK2-aggregates (Figure 5). In order to get a more robust conclusion, recombination markers should also be evaluated under these circumstances.

We have performed the experiment requested by the reviewer by performing DSB-2 staining before and after TEV-mediated removal of REC-8 in germlines that were previously depleted of CHK-2 activity using the auxin degron. As we previously observed with PLK-2 aggregates, removing REC-8 following CHK-2 depletion fails to induce reappearance of DSB-2 in pachytene nuclei. This result, shown in Figure S6C and mentioned in page 13 lines 15-16, confirms that CHK-2 is required for reimplementing pairing and recombination events of early prophase in pachytene nuclei following cohesin removal.

10-Next, and in order to determine if SC-disassembly-dependent re-entry into early meiosis is associated to transcriptional activity, the authors carry out RNAseq and conclude that, as expected from a CHK-2-dependent pathway, it is independent of new transcription. The western blot analysis of DSB-2 showing similar protein levels could also be carried out with an additional marker previously used (whenever technically possible) to demonstrate that protein levels are unchanged after depletion of cohesins REC8 and COH3.

The experiment suggested by the reviewer is not technically possible as there are no available antibodies to other recombination markers that display a staining pattern similar to DSB-2: they only associate with meiotic chromosomes during early prophase. SPO-11 would be a potential candidate for such experiment, but working antibodies against SPO-11 are not currently available.

11-In order to validate the conclusion that Cohesin-containing axial elements contribute to maintain CHK-2 activation (Figure 7), the SMC1 model that the authors have used previously in this work would validate it conclusively better than with the double COH3 and REC8.

We have performed the experiment requested by the reviewer by staining germlines of *smc-1::AID* worms treated with auxin for 16 hours, which induces full disassembly of axial elements in pachytene nuclei (Figure 2E), with PLK-2 antibodies. This experiment demonstrates that axis disassembly triggered by SMC-1 depletion prevents the formation of CHK-2-dependent PLK-2 aggregates on the nuclear envelope. Instead, most PLK-2 signals colocalise to a single aggregate of SC proteins present in the nucleus. These results, including a quantification of PLK-2 localization following 16 hours of SMC-1 depletion, are shown in Figure S5E and mentioned in page 16 lines 1-5. Importantly, as we did observe reappearance of PLK-2 aggregates in mid pachytene nuclei of *smc-1::AID::GFP* worms exposed to auxin for 16 hours, these results fully support our previous conclusion that cohesin-containing axial elements contribute to maintain CHK-2 activity.

Main formal comments to the MS

1-From my point the most controversial part of the MS is the title and the abstract. The title from my personal perspective could be drastically improved.

Regarding the abstract, it should be re-written for a more general reader. I would suggest to mention “the removal of the central element of the SC” instead of “removal of the SC” as the responsible of the phenotype observed. The SC is well referred to the tripartite structure but sometimes the axial elements or chromosome axes are also referred (non-orthodox way) as the growing SC. It should also be emphasized in the abstract the reversibility (re-entry into early prophase I) of the meiotic programme when the central element integrity is altered with different mutants (CE and cohesin axes). In general, the abstract of the paper does not summarize the experimental results that lead to the conclusions but mostly it is a list of conclusions in an abstracted way. I would suggest to summarize the experimental results that lead to a pair of clear general conclusions.

We have substantially modified the abstract following the suggestions of the reviewer and we have also modified our title to better summarize the main finding of our manuscript.

REVIEWERS' COMMENTS

Reviewer #1 (Remarks to the Author):

The authors have addressed my concerns.

Reviewer #2 (Remarks to the Author):

The authors had adequately addressed most of my suggestions. However, I still find the description of the quantification of nuclei in the germline to be rather obscure. I think it would help greatly if a sentence similar to what was provided in the author response was included in the main text on page 9 around line 28-29, before any quantification numbers are presented.

Reviewer #3 (Remarks to the Author):

All the concerns have been successfully addressed by the authors. I have no additional comments. The MS is of great interest and has been improved. The MS should be accepted now in the present form in Nature Communications.

Reviewer #1 (Remarks to the Author):

The authors have addressed my concerns.

We thank the reviewer for their support.

Reviewer #2 (Remarks to the Author):

The authors had adequately addressed most of my suggestions. However, I still find the description of the quantification of nuclei in the germline to be rather obscure. I think it would help greatly if a sentence similar to what was provided in the author response was included in the main text on page 9 around line 28-29, before any quantification numbers are presented.

As requested by the reviewer, we have introduced a sentence in page 9 to clarify the division of the germline into vertical rows of nuclei and we also direct the reader to Figure 1A, where a full description of this division is offered.

Reviewer #3 (Remarks to the Author):

All the concerns have been successfully addressed by the authors. I have no additional comments. The MS is of great interest and has been improved. The MS should be accepted now in the present form in Nature Communications.

We thank the reviewer for their support.